


**Timing landslide and flash flood events from SAR satellite: a new method illustrated in**
**African cloud-covered tropical environments**
Axel A.J. Deijns[1,2,*], Olivier Dewitte[1], Wim Thiery[2], Nicolas d'Oreye[3,4], Jean-Philippe Malet[5], François
Kervyn[1]
[1] Department of Earth Sciences, Royal Museum for Central Africa, 3080 Tervuren, Belgium
[2] Department of Hydrology and Hydraulic Engineering, Earth System Science, Vrije Universiteit Brussel,
1050 Elsene, Belgium
[3] Department of Geophysics/Astrophysics, National Museum of Natural History, 7256 Walferdange,
Luxembourg
[4] European Center for Geodynamics and Seismology, 7256 Walferdange, Luxembourg
[5] École et Observatoire des Sciences de la Terre & Institut Terre et Environnement de Strasbourg, Centre
National de la Recherche Scientifique, University of Strasbourg, F-67000 Strasbourg Cedex, France
[*] Corresponding Author. Email: axel.deijns@africamuseum.be
**Abstract**
Landslides and flash floods are geomorphic hazards (GH) that often co-occur and interact. They
generally occur very quickly, leading to catastrophic socioeconomic impacts. Understanding the
temporal patterns of occurrence of GH events is essential for hazard assessment, early warning and
disaster risk reduction strategies. However, temporal information is often poorly constrained, especially
in frequently cloud-covered tropical regions, where optical-based satellite data is insufficient. Here we
present a new method to accurately estimate GH event timing which requires no prior knowledge of the
GH event timing, using Synthetic Aperture Radar (SAR) remote sensing. SAR can penetrate through
clouds and therefore provides an ideal tool for constraining GH event timing. We use the open-access
Copernicus Sentinel-1 (S1) SAR satellite that provides global coverage, high spatial resolution (~10-15
m) and a high repeat time (6-12 days) from 2016 to 2020. We investigate the amplitude, detrended
amplitude, spatial amplitude correlation, coherence and detrended coherence time series in their
suitability to constrain GH event timing. We apply the method on four recent large GH events located
in Uganda, Rwanda, Burundi and DRC containing a total of about 2500 manually mapped landslides and
flash flood features located in several contrasting landscape types. The GH event timing estimation
accuracies vary among the GH events and the data products. Coherence and detrended coherence
estimated timing accuracies range from a 1 day to a 47 day difference. The spatial amplitude correlation
estimated timing accuracy ranges from a 1 day to an 85 day difference. The amplitude and detrended
amplitude estimated timing accuracies range from a 13 to a 1000 day difference. The amplitude time
series reflects the influence of seasonal dynamics, which causes the timing estimations to be further
away from the actual GH event occurrence compared to the other data products. Timing estimations
are generally closer to the actual GH event occurrence for GH events within homogenous densely
vegetated landscape, and further for GH events within complex cultivated heterogenous landscapes.
We believe that the complexity of the different contrasting landscapes we study is an added value for
the transferability of the method and together with the open access and global coverage of S1 data it
has the potential to be widely applicable.


## 1. Introduction

Landslides and flash floods are geomorphic hazards (GH) that can occur very quickly, sometimes in a matter of a few hours. GH frequently co-occur and interact (e.g. Rengers et al., 2016), they have a significant impact on the landscape (Petersen, 2001, Korup et al., 2010) and are severe threats for infrastructure and human life (Bradshaw et al., 2007, Kjekstad et al., 2009, Froude and Petley, 2018). For example, in 2013, several people were killed and ~7000 lost their homes in the Rwenzori Mountains in Uganda by a single debris-rich flash flood fed by upstream landslides (Jacobs et al., 2016a). Also, in 2011, a combination of flash flooding and mudslides across the highlands of the state of Rio de Janeiro claimed the lives of 916 people and left 35.000 people homeless (Marengo & Alves, 2012).

Understanding the temporal occurrence of GH events is essential for hazard assessment, early warning, and disaster risk reduction strategies (van Westen et al., 2008, Ali et al., 2017, Liu et al., 2018, Guzzetti et al. 2020). Temporal information with a few day accuracy is needed to understand the close association between precipitation and the occurrence of GH events (Guzetti et al., 2008; 2020, Turkington et al., 2014, Marc et al., 2018). For site-specific and local-scale investigation, this accurate information on the timing of GH events can be obtained with field-based approaches such as watershed/hillslope monitoring (Guzetti et al., 2012) or a network of observers (Jacobs et al., 2019). However, when information on the timing of GH events is needed at a regional level, the acquisition of such data can only be achieved with satellite remote sensing (Joyce et al., 2009, Le Cozannet et al., 2020), especially in mountainous regions with difficult field accessibility and where monitoring and observation capacities are limited (Dewitte et al., 2021).

Satellite remote sensing, and more specifically the use of optical imagery, is a well-developed field of research to accurately determine the location of GH (Stumpf et al., 2014, Behling et al., 2014; 2016, Mohan et al., 2021). Optical-based satellite approaches can also be used for extracting the information on the timing of the GH events (e.g. Kennedy et al., 2018, Deijns et al., 2020), however such approaches are of limited use in cloud-covered environments, especially if temporal information with a few day accuracy is needed.

Synthetic Aperture Radar (SAR) satellite, being an active system with an ability to penetrate cloud cover, holds a great potential for characterizing the timing of GH. Additionally, the sensitivity of SAR satellite data to surface changes, including vegetation changes (Hagberg et al., 1995, Balzter, 2001, Barrett et al., 2012), soil moisture changes (Dobson & Ulaby, 1986, Dubois et al., 1995, Ulaby et al., 1996, Nolan & Fatland, 2003, Srivastava et al., 2006), and surface texture changes (Dzurisin, 2006) gives SAR the potential to display GH timing with an accuracy of days.

GH events are usually analyzed using SAR amplitude data (i.e. changes in surface backscattering intensity of SAR signal between two images) (e.g. Mondini et al., 2017; 2019, Esposito et al., 2020, DeVries et al., 2020, Handwerger et al., 2022) for which amplitude correlation is a common method used in amplitude change detection (Mondini et al., 2017, Konishi & Suga, 2018, Jung and Yun et al.,





2020) or the interferometric coherence (i.e. the change in the ability of SAR wave fronts to stay spatially and/or temporally in phase between the two images of an interferometric pair) (Burrows et al., 2019; 2020, Tzouvaras et al., 2020). In recent studies, amplitude products are usually preferred over coherence products for GH detection (Ge et al., 2019, Jung and Yun et a., 2020, Mondini et al., 2021), since coherence generally yields less accurate results due to lower resolution (Burrows et al., 2019; 2020) and the higher number of false-positives (Aimaiti, 2019, Jung and Yun et al., 2020).

Despite the increasing use of SAR imagery for GH detection (Martinis et al., 2015, Twele et al., 2016, Mondini et al., 2019, Psomiadis et al., 2019, Burrows et al., 2020, Jacquemart and Tiampo, 2021, Jung and Yun, 2020, Tzouvaras et al., 2020, Handwerger et al., 2022), to date, only the recent study of Burrows et al. (2022) used SAR to refine the timing of GH inventories. Although located in the tropics and showing accurate results, their study was only applied (1) within a relatively densely vegetated landscape, (2) only on landslides, (3) using pre-processed amplitude imagery with Google Earth Engine (GEE) (Gorelick et al., 2017), (4) with a-priori knowledge on the timing of the event (i.e. the year) and (5) without consideration of the effect of vegetation dynamics within the timespan. Since GH events often occur on a regional scale (Emberson et al., 2020, Dewitte et al., 2021) there is a clear need to calibrate and validate any GH timing method for a variety of landscapes, and land use/land cover characteristics. Additionally, the frequent co-occurrence of landslides and flash floods (Jacobs et al., 2016b, Rengers et al., 2016) warrants the need to analyze them using a combined methodology.

The Copernicus Sentinel-1 (S1) constellation is frequently used in GH detection studies (Mondini et al., 2021). Next to the fact that it is freely available and acquired regionally (from 2016 onwards), it offers a very good trade-off between frequency of acquisition (6/12 days) and spatial resolution (10-15 m depending on the pre-processing parameters). These advantages make S1 an attractive tool to integrate in a regional GH timing methodology.

In this study, we aim to develop a method that automatically estimates GH event timing using S1 SAR imagery on GH events spatially located, but with unspecified timing. We create a method that can be applied at the regional scale in complex and various topographic and land use/land cover environments. Focusing on different landscape types observed in tropical Africa (see section 2.1), we analyze S1 SAR amplitude, spatial amplitude correlation (a metric based on the common amplitude correlation) and interferometric coherence changes. Specifically, we: (1) create S1 SAR time series and analyze their patterns and behavior at the location of several GH events, (2) demonstrate and assess the ability to detect the timing of GH events using changes within the S1 SAR time series and, (3) investigate the influences of the landscape characteristics on the ability to derive the timing from S1 SAR timeseries through a sensitivity analysis.



## 2. Data

### 2.1. Selection of GH events in a tropical region with diverse landscapes

We focus on the western branch of the East African Rift, a mountainous region with high population densities and diverse landscape and land use/land cover characteristics (Depicker et al., 2021a, Dewitte et al., 2021). The region has a bimodal precipitation distribution with two rainy peaks (October-November & March-April) and a main dry season (June-August) associated with the North-South migration of the Inter Tropical Convergence Zone (ITCZ) (Thiery et al., 2015, Nicholson 2017, Monsieurs et al., 2018a) with annual precipitation ranging from ~0.8m along the shores of lake Tanganyika to easily more than 2m in the highlands, with the maximum in the Rwenzori Mountains (Monsieurs et al., 2020, Van de Walle et al., 2020). The seasonality of the precipitation strongly controls the occurrence of landslides and flash floods (Jacobs et al., 2016a; 2016b, Monsieurs et al., 2018a; 2018b, Kubwimana et al., 2021). Vegetation dynamics are high in the cultivated areas due to the variety of cropping practices (crop rotations and shifting cultivation, Heri-Kazi & Bielders, 2021). Moreover, the region is one of the most cloud-covered places in the world (Robinson et al., 2019) and a global hotspot of thunderstorm activity (Thiery et al., 2016; 2017, Peterson et al., 2021).

We investigate four GH events with known days of occurrence, and located in contrasting landscapes (fig. 1):

- Event 1 (Uganda GH event) is located in the southern part of the Rwenzori Mountains (Uganda) and counts 1063 landslide features of which some contribute directly to the sediment load of the valley river (fig. 1, Uganda). The event occurred between the 21st and the 22nd of May 2020. The terrain consists of pristine forests and some cultivated landscape (fig. 2a).

- Event 2 (Rwanda GH event) is located in the Karongi district (western Province, Rwanda) and counts 494 features composed of both landslide and flash floods and occurred on the 6th of May 2018 (fig. 1, Rwanda). The terrain consists of an inhabited and highly cultivated landscape with the presence of agricultural terraces (fig. 2b).

- Event 3 (Burundi GH event) occurred around the hills of Nyempundu in the Cibitoke region (north Burundi) and counts 318 features composed of landslides and flash floods and occurred between the 4th and 5th of December 2019. Here, many landslides contribute directly to the sediment load of the rivers (fig. 1, Burundi). The terrain consists of inhabited cultivated landscape and sporadic tree cover (fig. 2c).

- Event 4 (DRC GH event) occurred west of the city of Uvira (DRC), northwest of Lake Tanganyika and counts 609 landslides and flash flood features that occurred between the 16th and the 17th of April 2020. Many landslides are connected to the rivers where the flash floods occurred. The debris-rich flash floods inundated parts of the city (fig. 1, DRC). The terrain is characterized by an urban area, cultivated landscape, grassland, and sporadic tree cover (fig. 2d).
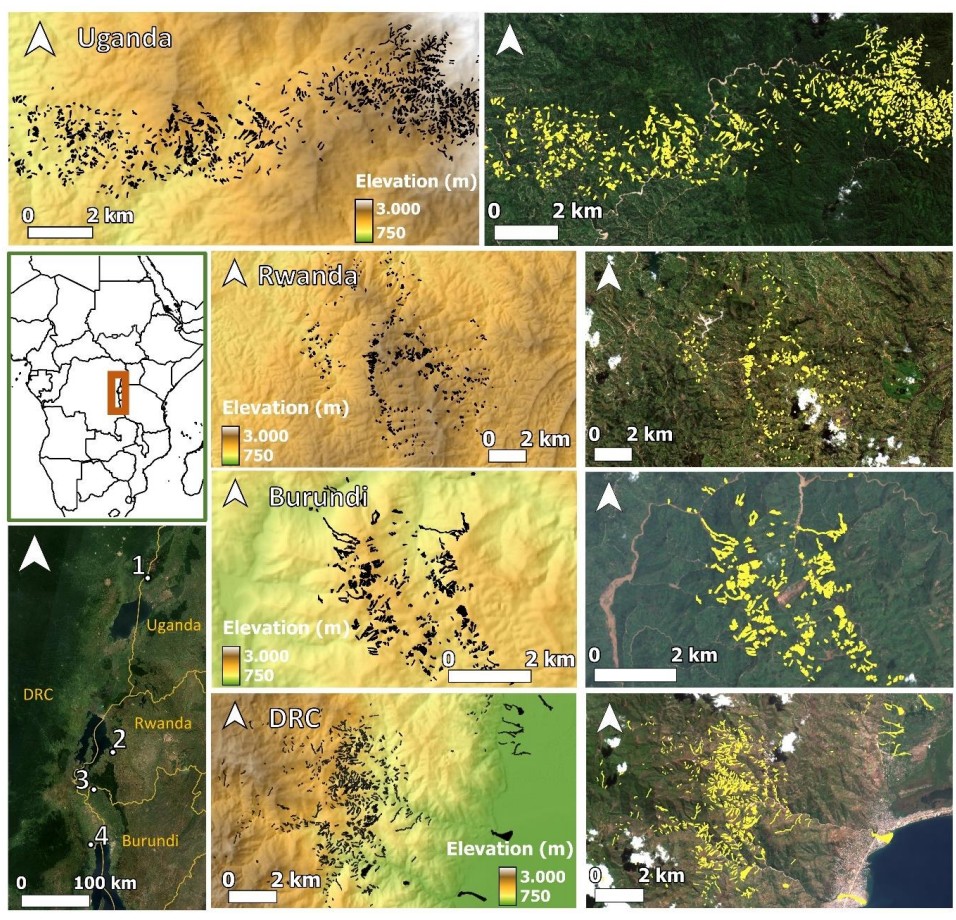

146

*Figure 1 The location of the four GH events with their topographic (left: 30m ALOS 3D DEM, GH event*
*features in black) and optical (right: S2 post-event image, GH event features in yellow) context. Note*
*that in the close vicinity of the GH events of Uganda and Burundi, large sediment-loaded riverbeds are*
*visible. This is a consequence of the GH events that contributed directly to the transport of extra material*
*to the rivers, increasing not only their sediment content, but also their lateral mobility. These river*
*dynamics are not included in our analysis. The two panels at the lower left depict the location of the GH*
*sites (S2 imagery). Image credit: Contains modified Copernicus Sentinel data (2022), processed with*
*© Google Earth Engine. ALOS 3D DEM data provided by Japan Aerospace Exploration Agency (JAXA).*

The locations of the GH events (fig. 1) are derived using the Copernicus Sentinel-2 (S2) Multispectral
Instrument (MSI), high resolution (10m), high frequency (6 -12 days) satellite imagery. We manually
digitized all individual events from the first available cloud-free S2 image after the event and a cloud-
free S2 image with similar vegetation characteristics (compared to the post-event image) before the
event. We use PlanetScope Ortho Scenes (Planet Team, 2017) for validation of the GH event inventory



with a higher resolution satellite image. Planet operates with a constellation of multiple small satellites
producing very-high resolution (3m), high frequency (up to 1 day) imagery (Table 1).
*Table 1: Images information of manual mapping and dating GH events. Planet images are of the type*
*PlanetScope Ortho Scene (POS)*

| GH Event | Sentinel-2 | | | | Planet | |
|---|---|---|---|---|---|---|
| | Date – pre | Date - post | Tile | Type | Date | Type |
| Uganda | 2019-08-16 | 2020-06-01 | 35NRA | L1C | 2020-06-29 | POS |
| Rwanda | 2018-03-09 | 2018-06-12 | 35MQT | L1C | 2019-12-07 | POS |
| Burundi | 2019-08-06 | 2020-01-23 | 35MQT | L1C | 2018-06-12 | POS |
| DRC | 2019-07-02 | 2020-06-06 | 35MQS | L1C | 2020-10-06 | POS |

We prefer the use of Planet and S2 over the Maxar or the Spot/Pléiades images visible in Google Earth
because of the consistency in temporal and spatial resolution. To note, the Burundi GH event has
recently been mapped by Emberson et al. (2022) by means of a semi-automated method followed by a
manual correction using S2 satellite data. We expect our manually mapped Burundi GH event inventory
to be similar or more accurate since we use a combination of S2 and Planet satellite data and a
completely manual detection workflow. The date of GH event occurrence is determined from local media
and field observations, and if not available from these resources, determined by the first- and last
available imagery from S2 and Planet imagery.

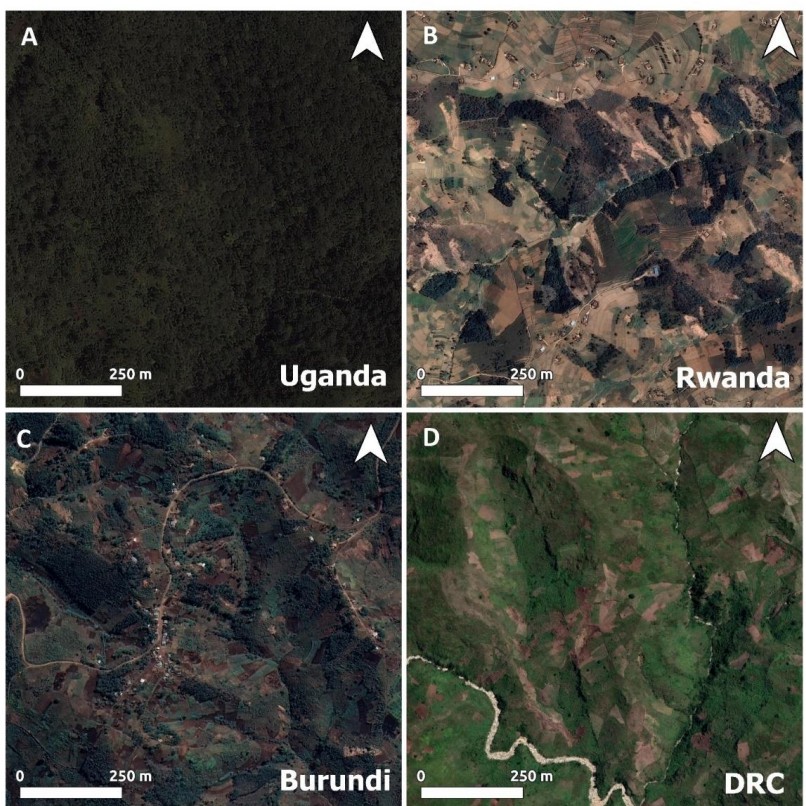

*Figure 2: Close up of the contrasting typical landscapes of the four GH events. Coordinates at image center (lat, lon): (a) 0.144°, 29,757°, (b) -2.171°, 29.410°, (c) -2.635°, 29.090°, (d) -3.339°, 29.119°. Maps Data: Google, ©2022 Maxar Technologies (a, c, d) Google, ©2022 CNES/Airbus (b).*

**2.2. SAR time series**

SAR time series at the GH location are constructed using the Copernicus S1 Level-1 Single Look Complex (SLC) imagery acquired in Interferometric Wideswath (IW). The S1 satellite is side-looking (right) and operates both on the ascending (from South to North) and descending (from North to South) tracks within the C-band frequency. To study the four GH events (fig. 1) we use between 196 and 208 ascending and 120 and 193 descending high resolution S1 images per GH event (~15x15 meter resolution) ranging from January 2016 up to January 2021 with a repeat time of six to twelve days with more consistently six days towards recent times. We use both amplitude and coherence information. S1 images over the study area are provided in vertical-vertical (VV) and vertical-horizontal (VH) polarizations. Different polarizations result in different backscattering values (Shibayama et al., 2015, Psomiadis, 2016, Park & Lee, 2019, Burrows et al., 2022). Mondini et al., 2019 noted a better definition of landslide-induced changes in vegetated areas using the VH channel. In contrast, Burrows et al. (2022)

found VV to perform better than VH for landslide event timing estimation. Psomiadis (2016) concluded
that VV polarization performed better than VH polarization for flash flood mapping. Finally, VV
polarization images are acquired more consistently at the locations of our GH events. We therefore
decide to use VV polarization for our analysis. Due to the side-looking nature of the S1 satellite it is
subjected to foreshortening, layover, and shadowing which are SAR inherent quality problems that are
amplified within mountainous regions and affect image quality (Hanssen, 2001, Dzurisin, 2006). GH
inventories are masked for foreshortening, layover, and shadow areas to remove the individual
landslides and flash floods that fall within these inherently noisy areas.

**2.3. SAR controlling factors**

SAR amplitude and coherence are influenced by local slope angle (Hanssen 2001), soil moisture (Ulaby
et al., 1996, Scott et al., 2017), vegetation (Balzter, 2001, Barrett et al., 2012), and terrain roughness
(Dzurisin, 2006). Coherence is additionally influenced by atmospheric changes (Rocca et al., 2000) and
due to the use of image pairs, also by the temporal baseline (time between acquisition of two images),
the perpendicular baseline (distance between the location of acquisition of two images) and the
difference in incident angle of the paired images (Hanssen, 2001). Coherence values are generally very
low (high decorrelation) in densely forested areas due to constant movement of the leaves and stems
(Weydahl, 2001, Tessari et al., 2017), whereas bare soils or urbanized terrains, due to their static
nature, generally reveal relatively high coherence values (Colesanti & Wasowski, 2006). An increase in
coherence values after GH event occurrence is therefore expected. Amplitude values, on the other hand,
show to have a quite complex reaction to terrain change. Due to the influence of soil moisture and
roughness change on the amplitude values, the occurrence of a GH event could both increase and
decrease the amplitude values at the location of the GH event (Mondini et al., 2021, Burrows et al.,
2022). Both precipitation (in changing leaf- and soil wetness) and vegetation patterns, can dynamically
influence SAR  amplitude and coherence values, causing a cumulative effect on the time series
(Srivastava et al., 2006, Brancato et al., 2017). This effect is more prominent over sparsely vegetated
areas due to geometric (vegetation growth and farming practices) and dielectric (moisture) changes
(Strozzi et al., 2000). Additionally, a change in atmosphere (precipitation events, ionospheric
disturbances) can dynamically influence the coherence values (Rocca et al., 2000, Jacquemart &
Tiampo, 2021). To better assess the ability to detect GH timing, it is essential to understand the dynamic
factors controlling the behavior of the signal.
We derive precipitation estimates from the GPM Level 3 IMERG Final Daily (10km spatial resolution)
dataset that has been validated through rain gauge data within the area (Nakulopa et al. 2022). General
vegetation patterns per GH event are visualized using the Normalized Difference Vegetation Index
(NDVI; Tucker, 1979). NDVI time series are derived from the Landsat-8 (30m spatial resolution) archive
and processed within the GEE environment We use the Landsat 8 atmospherically corrected surface
reflectance images provided within the GEE environment. We masked them for clouds using the quality
assessment band resulting from the CFmask algorithm (Foga et al., 2017).
We choose the lower resolution Landsat-8 over the higher resolution S2 imagery to reduce any
unwanted local effects of NDVI change captured in the higher resolution S2 imagery, and since we are
only interested in the general vegetation trends within the area this should be sufficient. From the cloud-
masked images, a spatial-average NDVI time series is created spanning from 2016-2020 over the
undisturbed areas of the GH event area. The NDVI time series are further processed to monthly
averages, since we are interested in general vegetation patterns visible in the NDVI time series rather
than changes on smaller temporal timescales.
We use the ESA Climate Change Initiative Land Cover product (ESA, 2016) to categorize GH based on
their prior land cover to assess the influence of land cover on the timing detectability. This product has
been validated within the region by Depicker et al. (2021), showing an accuracy of 86.1 ± 2.1% in land
cover classification. All above mentioned factors are considered during the analysis of the SAR timeseries
and the GH event timing estimations.

### 3. Methods

### 3.1. Sentinel-1 pre-processing

The S1 images are pre-processed using the "InSAR automated Mass processing Toolbox for
Multidimensional time series" (MasTer) (Derauw et al., 2020, d'Oreye et al., 2021) processing chain (fig.
3, step 1). MasTer is a tool for automated SAR and SAR interferometry (InSAR) mass processing
(Samsonov & d'Oreye, 2012, Derauw et al., 2019; 2020, d'Oreye et al., 2019; 2021), that is incremental
(i.e. only computes the minimal required information when a new image is available) and optimized for
mass processing. The MasTer workflow is applied on both the ascending and descending track and
consists of:
(1) the application of orbit correction using the precise orbit files provided with the S1 data.
(2) The creation of time series of amplitude maps per track. Amplitude maps of each given track are
co-registered on a reference image taken from that track. Every amplitude image in the radar geometry
of that track is cropped and provided with the same grid and dimensions framing the area of interest.
Amplitude values are calibrated to sigma nought values. The amplitude images are multi-looked by a
factor 2 in azimuth and in range, leading to a roughly 28x5 m slant range resolution. Radiometric terrain
correction is applied to account for the local incidence angle variating with slope angle resulting in
amplitude values that are independent of slope angle (Small, 2011).
(3) The creation of coherence maps using consecutive images throughout the time series with a
maximum temporal baseline of 12 days and a maximum perpendicular baseline of 150 m. The coherence
maps are provided with the same multi-looking factor, grid, and ground range resolution as the
amplitude images.
(4) All the amplitude and coherence maps from all the tracks spanning a given GH area are geocoded
from slant range to ground range on a common grid with a 15 by 15 m resolution using the 30 m ALOS


Global Digital Surface Model. We decided to geocode the SAR imagery to make it compatible with all
our other data products and to allow for an easier visual comparison with optical imagery.

## 3.2. Spatial amplitude correlation

We adapt the amplitude correlation approach, initially used for GH spatial detection (Mondini et al.,
2017, Konishi & Suga, 2018, Jung & Yun, 2020), to allow for GH timing detection at the location of the
GH event using the amplitude image stacks (fig. 3, part 2). We reason that the spatial correlation is
generally lost when the inter-pixel relationships between two images change at the location of a GH
event. Therefore, a significant change within the landscape such as a landslide or a flash flood will cause
a significant decorrelation. Due to the sensitivity of SAR amplitude to changes in vegetation (Balzter,
2001, Barrett et al., 2012), seasonal greening and browning trends have a pronounced influence on the
amplitude time series (Balzter, 2001, Barrett et al., 2012), which potentially limits the detectability of
the GH event within the time series. Since spatial correlation is only changing when the inter-pixel
relationships change, general trends that affect the entire area (lowering or increasing the SAR
amplitude values) do not influence the inter-pixel relationships (i.e. no spatial correlation change). Only
when significant inter-pixel change occurs, due to landslides or flash floods, the spatial correlation will
change. The spatial amplitude correlation (SAC) can therefore highlight the GH event occurrence within
the time series, while reducing the seasonal dynamics. To calculate the SAC, we use equation 1 that we
adapted from Jung & Yun (2020).
$$SAC_{x,y,poly} = \frac{\sum\{(A_{r,poly} - \overline{A_{r,poly}})(A_{x,poly} - \overline{A_{x,poly}})\}}{\sqrt{\sum\{(A_{r,poly} - \overline{A_{r,poly}})^2\}\sum\{(A_{x,poly} - \overline{A_{x,poly}})^2\}}} \quad x = date_1 \dots date_{N+1}; x \neq r \qquad (1)$$
with $SAC_{x,r,poly}$ the spatial amplitude correlation for the impacted area (timing workflow 1: The complete
GH event, timing workflow 2: per individual GH feature) of date x in reference to date r, $A_{x,\,poly}$ the
amplitude pixels of impacted area at date x, and $A_{r,\,poly}$ the amplitude pixels of impacted area at reference
date r. Instead of calculating correlation between two subsequent images over a given window, we
calculate the correlation using one reference image ($A_r$) and all the other images within the time series
($A_x$) using only the pixels within a designated impacted area (e.g. single GH feature or complete GH
event) ($_{poly}$). Consequently, every image within the amplitude image stack can be used as a reference
image and due to slight changes within every amplitude image this will inevitably result in different SAC
time series, one better highlighting the GH event than the other. We apply the equation separately for
ascending and descending images in a parallel workflow. Figure 4 shows schematically how the SAC
time series should behave using different reference images. Taking a reference amplitude image before
the GH event occurrence (fig 4a), results in high SAC before and low SAC after GH event occurrence.
The opposite is expected when using a reference amplitude image after the GH event (fig 4b).
We use every available image within the amplitude image stack as a reference image and calculated
the respective SAC time series from it. From here, it is necessary to identify the most appropriate
reference image.


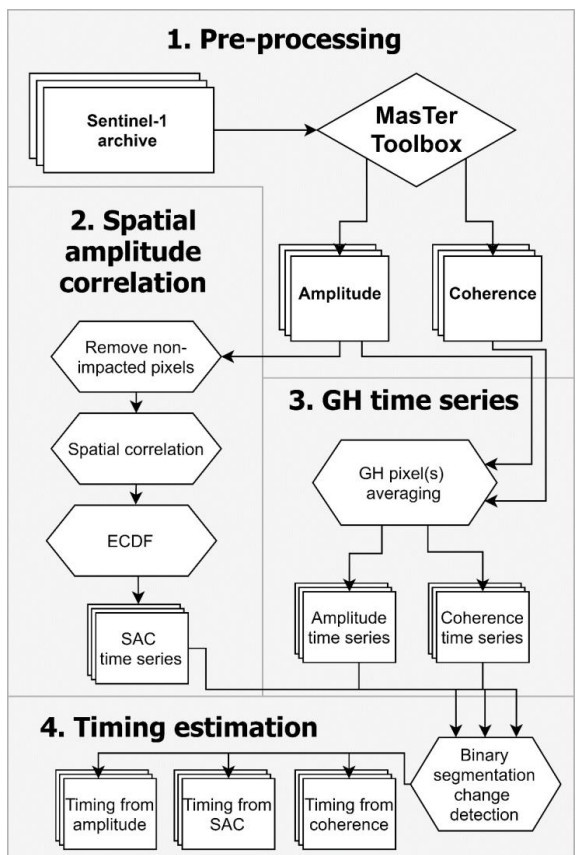


Figure 3: Workflow of the methodology. Rectangles represent initial input imagery, output image stacks
or time series products. The rhombus represents the external software product. Hexagons represent
methodological steps, which are described in the text. (1) Pre-processing of the S1 imagery using the
MasTer processing chain to acquire amplitude and coherence image stacks. (2) Application of the spatial
amplitude correlation (SAC) method using Empirical Cumulative Distibution Functions (ECDF) on the
amplitude image stack resulting into SAC time series. (3) GH pixel(s) averaging for every image in the
amplitude and coherence image stacks resulting into amplitude and coherence time series. (4)
Application of binary segmentation change detection to acquire the date of the most significant change
within the amplitude, SAC, and coherence time series.

Hence, we develop a new method that identifies the most suitable reference amplitude image by finding
the SAC time series that most distinctively shows changes related to the GH event occurrence. We
distribute every SAC time series as Empirical Cumulative Distribution Functions (ECDF) resulting in
multiple ECDF curves equal to the amount of reference images. A SAC time series that contains a distinct
change indicative of the GH event occurrence will show a similar distinct change in its ECDF.
Contrastingly, SAC time series that fail to distinctively highlight the GH event, show an ECDF that is
similar to a normally distributed ECDF. Therefore, we create a normally distributed ECDF, using the





mean and standard deviation derived from the ensemble of ECDF curves, and identify the ECDF that
deviates most from it. Per ECDF we calculate and cumulate the difference from the normally distributed
ECDF. The ECDF with the highest cumulative difference is chosen as most representative and the related
SAC time series was used.

**3.3. GH event timing estimation**

GH event timing is determined on two scales within separate workflows:
Timing workflow 1: the complete GH event scale. This workflow contains all pixels encompassing the
full GH event, resulting in an ascending and descending track time series for amplitude, SAC, and
coherence.
Timing workflow 2: the individual GH scale. In this workflow, the GH event is subdivided in multiple
individual GH features, resulting in multiple ascending and descending track time series, equal to the
amount of individual GH features, for amplitude, SAC, and coherence.
In both workflows we do not choose to remove fuzzy pixels (i.e., edge pixels that contain both impacted
and non-impacted landscape). Since we do not know the effect of these pixels on the SAR time series
and GH event timing estimations, we apply the analyses without additional processing of the GH event
inventories. This allows us to establish baseline results. The ascending and descending track data are
processed separately throughout the two workflows. Amplitude and coherence time series are generated
by averaging the values within the identified impacted area per image (fig 3.3) and the SAC time series
are generated by applying the SAC method (fig 3.2, section 3.2) on the same area (workflow 1: the
complete GH event, timing workflow 2: per individual GH feature). The resulting time series are
normalized using the time series average to improve comparability.
Additionally, we make an effort to remove the seasonal influence and atmospheric effect on the
amplitude and coherence time series by subtracting the regional amplitude and coherence trend (i.e.,
time series) from the GH event scale amplitude and coherence time series (timing workflow 1). Both
precipitation events and seasonal vegetation dynamics are expected to cover the complete GH event
and its surrounding area. This detrending will therefore emphasize the change induced by the GH event
occurrence while removing any regional changes induced by either seasonal vegetation dynamics or
atmospheric effects (e.g. Jacquemart & Tiampo, 2021). The regional amplitude and coherence time
series are established by following sections 1 and 3 from the methodology (fig. 3), using a larger area
surrounding the GH events as input (i.e. a square of approx. 1.5 times the GH event area, excluding
the exact location of the GH event). This results in the detrended amplitude and detrended coherence
data products. Given the fact that SAC is based on inter-pixel changes, subtracting a general value as
a mean of detrending would make no difference. Moreover, SAC is created to already consider seasonal
vegetation dynamics so no additional detrending was needed.




We decide not to detrend individual GH feature time series (timing workflow 2), which would include
the use of a detrending buffer such as in Burrows et al. (2022). Since we deal with complex
heterogenous land cover, proximate landscape does not necessarily represent the landscape at the
individual GH feature. Additional research would therefore be required to accurately implement such a
detrending method that is expected to be applicable in a wide variety of environments.
Timing is then defined on every time series using a binary segmentation change detection approach
(Bai, 1997, Fryzlewicz, 2014) using ruptures (Truong et al., 2020). This allows us to locate, in time, the
largest change within every time series. On the complete GH event scale (timing workflow 1) this results
in two dates (from ascending and descending track) per data product (amplitude, detrended amplitude,
SAC, coherence, detrended coherence). On the individual GH scale (timing workflow 2), this results in
several dates, equal to two times (one for ascending and one for descending track) the amount of
individual GH features per data product (amplitude, SAC, coherence). Here we identify the date  that
occurred most frequently (majority) as representing the timing of the event.
We expect that the coherence image pair that demonstrates an increase in coherence compared to the
former coherence image pair consists of less vegetated terrain (as caused by the GH event) and thus
contains post-event conditions (Tzouvaras et al., 2020, Burrows et al., 2020; 2021). The first date from
this post-event coherence image pair is therefore extracted and defined as representing the timing of
the event. We define the minimal uncertainty in timing estimation by the difference between the
estimated image date and the date of the image before that one within the image stack (maximum of
12 days).

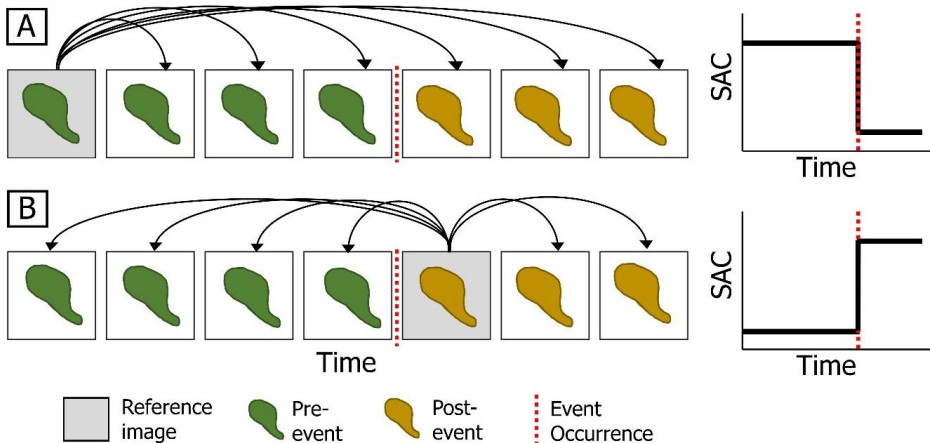


*Figure 4: idealized scheme of the SAC method using 2 different reference images: one before and one*
*after the occurrence of the GH event (A, B). Squares represent images, the red dotted line indicates the*
*occurrence of a GH event. Inside the images are the conditions of the impacted area (represented here*
*as single GH feature but is similar for complete GH event). Pre-event conditions are displayed in green.*




*Post-event conditions are displayed in brown. The black curved lines represent the combination of*
*images on which equation 1 is applied to achieve the resulting SAC time series. The schematic SAC*
*graphs (right) depict the expected results using a reference image before the event (A) with high*
*correlation before and low correlation after the event, and using a reference image after the event (B)*
*with low correlation before and high correlation after the event.*

### 3.4. Time series analysis

In section 2.3 we discuss the controlling factors on the SAR signal. Here, we try to understand the
influence of these controlling factors plus the influence of individual GH properties on the detectability
of the event timing. We carry out a sensitivity analysis on GH area (effect of a changing number of
pixels/pixel mixing, Deijns et al., 2020), slope angle (change in image acquisition geometry, Zebker and
Villasenor, 1992,  Hanssen, 2001), land cover (changing vegetation and soil moisture patterns, Giertz
et al., 2005), and slope aspect (different effect of layover, shadowing within ascending and descending
track, Hanssen, 2001, Dzurisin, 2006). We carry out the analysis separately for the ascending and
descending track images. Per individual GH feature we derive the average value of the above-mentioned
parameters. We find more smaller-sized GH in the Rwanda GH event (fig 5a), a slight deviation (peak
more to the left) in slope distribution for the Uganda GH event (fig. 5b) and a large variation in slope
aspect distribution for different GH events (fig. 5d). Additionally, land cover distribution is different for
every GH event (fig. 5c) which corroborates with what we see on the satellite images (fig. 2).
The sensitivity analysis is carried out iteratively over every parameter from a minimum value to a
maximum value using predefined steps (Area: 1000 m$^2$, Slope: 5°, Land Cover: per individual land cover
type, Slope aspect: 45°). Per iteration the GH inventory is reduced to contain only individual GH features
that meet the iteration conditions. We exclude bins that contained less than 20 individual GH features
to avoid non-sense (very high or very low) values that would negatively influence the quality of the
trend.
Per bin-size, the timing is calculated for every individual GH feature, and the percentage of timing
estimates that fall within one month of the actual event occurrence over the total amount of individual
GH features within that specific bin is calculated. Higher percentages indicate more timing estimates
closer to the actual event occurrence. The variations within this percentage are subsequently analyzed
to relate changing characteristic to performance.

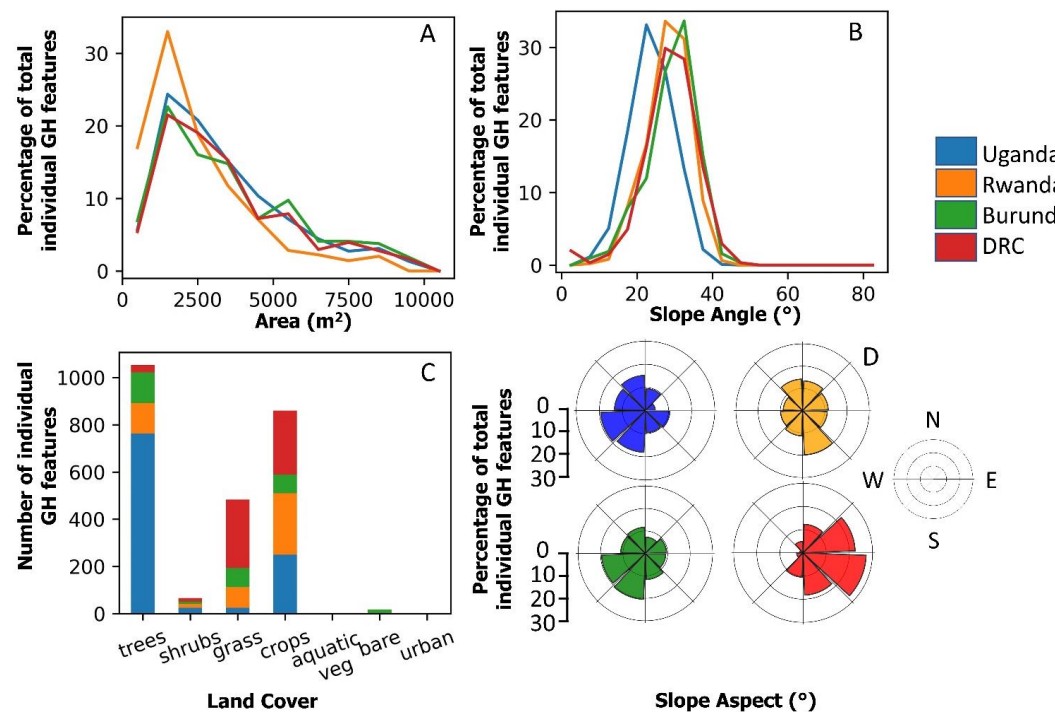

Figure 5: parameter distributions per GH event (Uganda, Rwanda, Burundi, and DRC). (A) Percentage of individual GH over total amount of individual GH against area (m²), bins of 1000 m². (B) Percentage of individual GH over total amount of individual GH against slope angle, bins of 5°. (C) Number of individual GH against land use/land cover. (D) Percentage of individual GH over total amount of individual GH against slope aspect, bins of 15°.



## 4. Results

### 4.1. GH event time series

We created amplitude, detrended amplitude, SAC, coherence, detrended coherence time series for the four GH events in Uganda, Rwanda, Burundi, and DRC (location in fig. 1) and present it in figure 6 together with the average monthly Landsat 8 NDVI and IMERG monthly cumulative precipitation.

The distinctiveness of the GH event occurrence within the time series varies significantly per data product (fig. 6). SAC (fig. 6i-l) and coherence (fig. 6m-t) time series showcase the timing of the event with a significant change of value at the time of the event occurrence. Change in coherence due to the GH event is clearly indicated by the increase in value starting from the post-event coherence pair. A significant decrease for the co-event (the coherence value from the pre- and post-event image) coherence pair is not visible.

The amplitude time series do not show any distinct change at the time of the GH event occurrence (fig. 6 a-h), except for the Uganda GH event (fig 6a,e). Particularly in the amplitude time series, and to a minor extent in the coherence time series, clear cyclicity can be observed, which corresponds with the two drier periods (December-February and June-August) that are prevalent in the region (Bonfils, 2012, Nicholson 2017, Monsieurs, 2018a). The NDVI shows seasonal correlation with the precipitation patterns, where NDVI patterns follow precipitation patterns with a short time lag (fig 6. u-x). Stronger NDVI variations align with a stronger cyclicity within the amplitude, SAC, and coherence time series which is particularly visible when comparing the Uganda GH event (weak amplitude SAC and coherence cyclicity, limited NDVI fluctuations) and the DRC GH event (stronger amplitude, SAC, and coherence cyclicity, large NDVI fluctuations). When comparing the landscape of both GH events (fig. 2a,d) a sharp contrast is observed. The Uganda GH event region is mostly covered by forest, whereas the DRC GH event region is mostly covered by grass- and cropland. Consequently, we find that seasonal NDVI oscillations vary significantly from one study area to another given the difference in landscape. The seasonal oscillations in vegetation are visible within the amplitude and coherence timeseries and subsequently influence the distinctiveness of the GH event within these timeseries.

Time series detrending clearly reduces seasonal cyclicity within the time series, which is particularly visible for the coherence time series (fig. 6q-t) and to a much smaller degree for the amplitude time series (fig. 6e-h). For example, the DRC GH event coherence time series benefits from this detrending procedure such that seasonal cyclicity is almost completely removed and in the resulting time series a sudden increase in coherence values can be observed after the occurrence of the GH event (fig. 6t). Detrending of the amplitude shows some improvements but is remains difficult to the GH event within the time series remains poor.

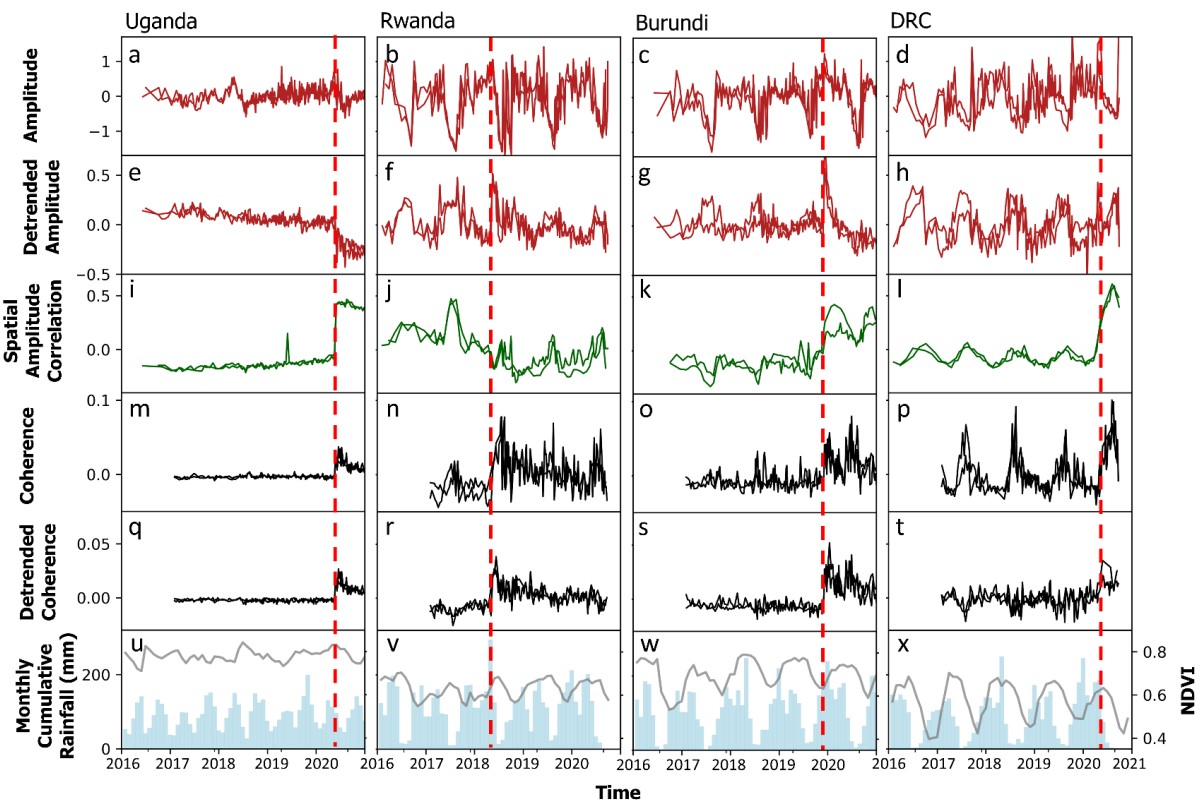

*Figure 6: GH event (detrended) amplitude (red), spatial amplitude correlation (SAC, green) and*
*(detrended) coherence (black) time series. The dashed red line represents the timing of the GH event*
*occurrence within the time series. All coherence, amplitude and SAC time series show two lines of a*
*similar color representing the ascending and descending track time series. The time series are created*
*according to the complete GH event scale method described in sections 3.2 (SAC) & 3.3 (amplitude and*
*coherence). The bottom row shows the monthly cumulative precipitation (light blue bars) from IMERG*
*satellite data and the monthly averaged NDVI values (grey line) from Landsat 8 (method described in*
*section 2.3).*

**4.2. GH event timing**

Figure 7 shows the timing estimation at the GH event scale (timing workflow 1) from the (detrended) amplitude, SAC and (detrended) coherence time series. The timing range (i.e. uncertainty in estimated timing, length of the bars in fig. 7) is defined by the minimum and maximum difference in days of the estimated timing from the actual GH event occurrence and takes into account the estimated pre-event image (or image pair for coherence), estimated post-event image (or image pair for coherence) and the actual GH event occurrence timing range. Estimation from the amplitude time series perform poorly



with estimated timing ranging from a 46 day difference (Uganda, descending) to a 1000 day difference
(Uganda, ascending). Estimations from the SAC time series range between a 1 day (Uganda) and an 85
day (Rwanda) difference and estimations from the coherence time series range between a 1 day
(Uganda) and a 47 day (Rwanda) difference. Highest accuracies are achieved with times series showing
less seasonal fluctuation and a steep change at the time of event occurrence (fig. 5). Timing estimations
from the detrended amplitude time series show increased accuracy compared to amplitude time series
with the most significant change for the Uganda GH event from a 46-1000 to a 13-22 day difference,
but performance is still poor and generally useless for accurate timing estimation. Detrending the
coherence time series increases timing estimation accuracy compared to the non-detrended coherence
timing estimation for the DRC event (25-32 to a 1-5 day difference), but in general the estimations
remain the same.
Figure 8 shows the timing estimation based on the individual GH features within the GH event (timing
workflow 2). Similar to figure 7, timing range (i.e. uncertainty in estimated timing, length of the bars in
fig. 8) is defined by the minimum and maximum difference in days of the estimated timing from the
actual GH event occurrence. Here, the estimated timing represents the date that is estimated most
frequently between all individual GH features (as explained in section 3.3). The percentage of individual
GH features that estimate this (most frequently estimated) date over the total amount of GH features
(%maj) is included in figure 8.
In general, timing estimation from the amplitude time series performs rather poorly with estimated
timing ranging from a 13 day difference to an 831 day difference. A distinct increase in estimated timing
accuracy, compared to the results from the GH event scale (fig. 7, timing workflow 1), is seen for the
Uganda GH event. But the other GH events do not show any distinct increase in timing estimation
accuracy. The %maj ranges between 13 and 32.4 and shows that for some GH events a large portion
of the individual GH features estimate a date that is far from the actual date of the GH event occurrence.
The percentage of individual GH features that estimate a date within one month of the actual GH event
occurrence from amplitude time series is 24.2% (ascending) and 26.9% (descending) for the Uganda
GH event, but much lower for the other GH events, corroborating the fact that the timing detection
method performs poorly with the amplitude data product.
Timing estimations from the SAC time series from individual GH features (fig. 8, timing workflow 2)
differ compared to the timing estimations at the GH event scale (fig. 7, timing workflow 1). An increase
in accuracy is seen for Rwanda (ascending) and DRC (ascending) and a decrease in accuracy for Burundi
(ascending) and DRC (descending). The estimated timing ranges from a 1 day difference to an 85 day
difference. Although estimated timing accuracy is higher for SAC compared to amplitude, %maj values
are quite low, indicating weak estimations. The percentage of individual GH that estimate a date within
one month of the actual GH event occurrence ranges from 0.2 (Rwanda, descending) to 38,1 (Uganda,
descending). Exceptionally, for the Uganda GH event, %maj and estimated timing within one month of



the GH event occurrence from the SAC time series is highest in comparison with amplitude and
coherence (fig. 8).
Timing estimations from the coherence time series from individual GH features (fig. 8) are similar to
those achieved at the GH event scale (fig. 7), and have, generally, the highest accuracy for all data
products. The %maj values ranged from 13.5 (Burundi, ascending) to 38.4 (DRC, descending). The
percentage of individual GH features that estimates a date within one month of the actual GH event
occurrence ranges from 0 (Rwanda, descending) to 38,4 (DRC, descending). The low percentages from
the Rwanda descending track can be attributed to the fact that the estimated date is 37 days from the
GH event occurrence and therefore just falls outside the one-month threshold.

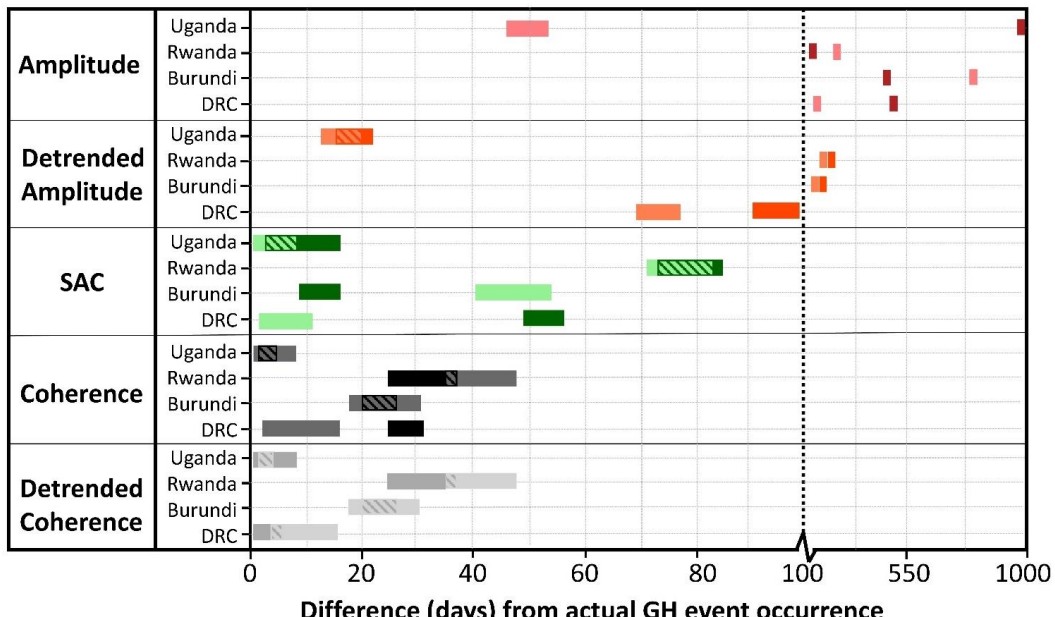


*Figure 7: Estimated GH event timing using the complete GH event scale. Per GH event two bars indicate*
*the estimated timing. The darker color bar visualizes the timing range estimated from ascending track*
*imagery and the lighter color bar visualizes the timing range estimated from descending track imagery.*
*The color dashed bar (▨) represents the overlap between ascending and descending track timing*
*estimations.*

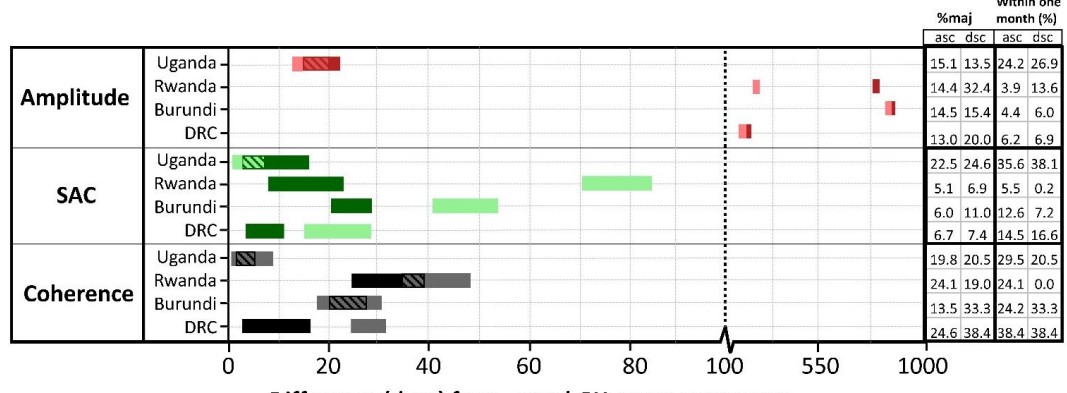


*Figure 8: Estimated timing from the individual GH scale. The bars represent the uncertainty in timing.*


*Per GH event two bars indicate the estimated timing. The darker color bar visualizes the timing range*


*estimated from ascending track imagery and the lighter color bar visualizes the timing range estimated*


*from descending track imagery. The color dashed bar (▨) represents the overlap between ascending*


*and descending track timing estimations. In the %maj column we present the percentage of individual*


*GH features over the total amount of individual GH features that were included in the majority vote*


*separated for the ascending (asc) and descending (dsc) track. In the 'within one month' column we*


*present the percentage of individual GH features over the total amount of individual GH features that*


*estimated a date within one month of the actual event occurrence.*


### 4.3. Time series analysis


GH size seems to have a clear influence on estimation accuracy. Specifically, the SAC and coherence


show a clear increase in percentages of estimated timing within one month of the GH event occurrence


with increasing GH size (fig. 9 a-f). $R^2$ values show a relatively reliable fit for both SAC and coherence.


Amplitude shows a slight increasing trend, but associated $R^2$ values are non-reliable.


Slope trend lines (fig. 9 g-l) show in general little to no inclination and $R^2$-values are insignificant, except


for the coherence ascending track. Here, a clear increase in slope angle becomes visible with a


comparatively higher $R^2$ (although clearly less reliable than $R^2$ from the area analysis).


To assess the influence of land cover we combined both the ascending and descending track results for


all four GH events in each boxplot (fig. 9 m-o). Each boxplot therefore contains a total of eight data


sources per land cover type. The major land cover classes within the GH events were tree covered area,


grassland, and cropland (Fig 4d). Median percentage values range around 9-10 % for amplitude, 11-16


% for SAC, and 27-34 % for coherence. Although median values within the grassland land cover type


seem to be systematically higher among the 3 data products (amplitude, SAC, and coherence),


differences with other land covers are quite small. No specific land cover shows a significant advantage.

To assess the influence of the slope orientation, we derive the difference between ascending and
descending track percentages per bin and determine which track shows better performance (fig. 9p-s).
At the results for the Rwanda GH event (fig. 9q) we see for SAC and coherence an all-round favorability
for the ascending track, that can be explained by the fact that, like the results in figure 8, the Rwanda
GH event had almost no estimations within one month of the GH event occurrence for the descending
track. The results presented for Uganda, Burundi and DRC GH events (fig. 9p,r,s) show a general
favorability of the ascending track for individual GH features that have an aspect of approximately 45-
180°, whereas a general favorability of the descending track for individual GH features that have an
aspect of approximately 225-360°. In contrast to this general trend, the opposite seems to be visible
for the Uganda GH event coherence.

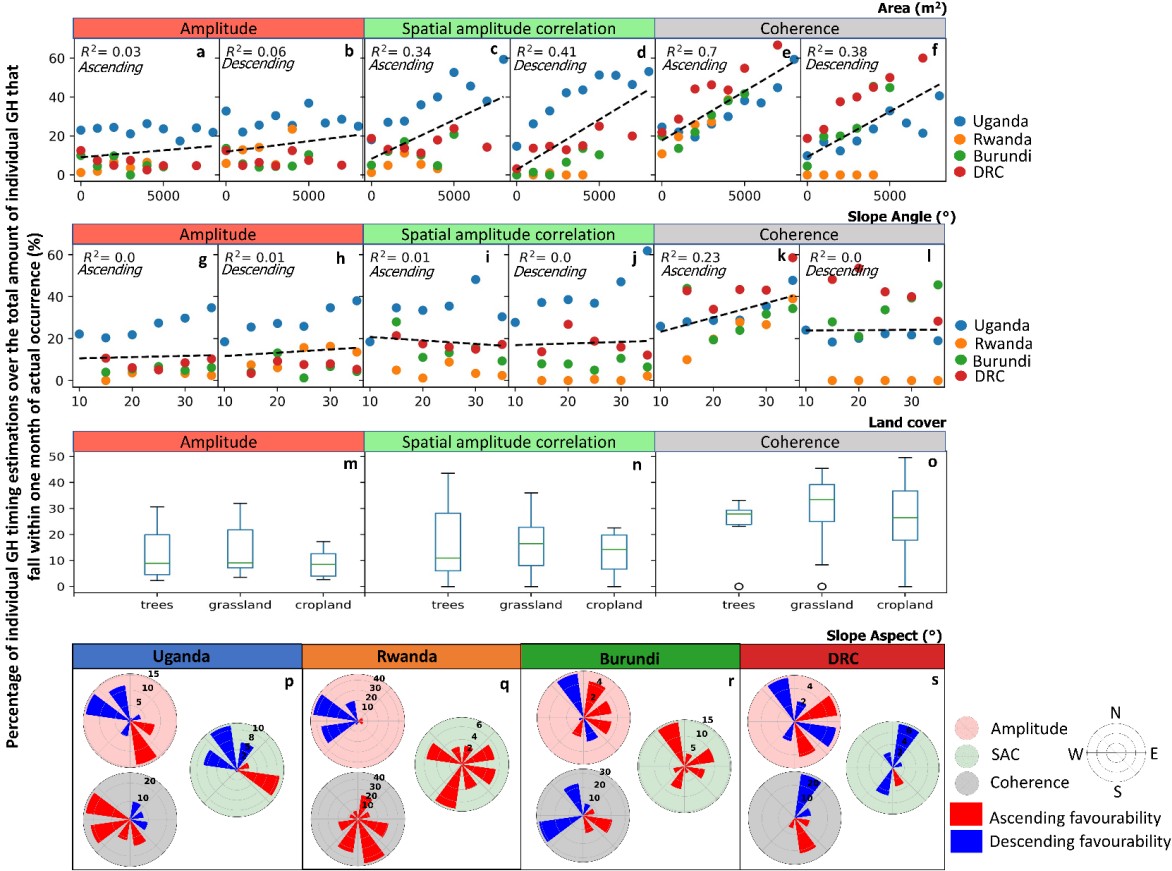


*Figure 9: Timing estimation performance over changing individual GH feature area (a-f), slope angle (g-*

*l), land cover (m-o) and slope aspect (p-s). The y-axis displays the percentage of individual GH features*

*that estimated a timing that falls within one month of the actual GH event occurrence over the total*





*amount of individual GH features per GH event. Bin sizes: area=1000m, slope angle=5°, slope*
*aspect=45°. Area (a-f) and slope angle (g-l) plots are separated per track, and the colors indicate the*
*different GH events. The black dashed lines present the linear trend lines fitted to the data (a-l) for*
*which the associated $R^2$ values are included. Land cover (m-o): boxplots give lower and upper quartiles*
*and median. The whiskers of each box represent 1.5 times the interquartile range. Outliers beyond*
*whiskers are shown as dots. Slope aspect (p-s): the polar plots present the favorability of the ascending*
*(ASC) or descending (DSC) track per slope aspect (see section 4.3). The color of the polar plot*
*background indicates the SAR data product.*
**5 Discussion**
In this study we present a methodology to automatically determine GH event timing using S1 SAR data.
Our study improves on the recent advances in GH event timing estimation research as: (1) we are one
of the firsts to use amplitude, SAC and coherence time series in a systematic manner to detect the
timing of GH events (Mondini et al., 2021), (2) we defined a methodology where no prior knowledge of
the GH event timing is required, (3) we applied our method on contrasting landscapes and (4) we
combined, for the first time, landslides and flash floods in a single detection approach. Here we discuss
our insights, results considering recent developments, and the potential improvements and future
perspectives of our new method.
**5.1. Insights in GH event timing estimation from SAR**
**5.1.1 GH event timing estimation**
The use of amplitude or detrended amplitude time series in our methodology does not prove to be an
effective approach to accurately determine the timing of GH events since it gives an estimation accuracy
of 13 to 1000 days with the actual time occurrence of the events. A clear increase in accuracy is obtained
from SAC with an accuracy of 1 to 85 day. However, the most accurate results are achieved with
coherence and detrended coherence with a 1 to a 47 day accuracy.
GH event timing accuracies are higher for GH events that occurred in remote areas with low amounts
of cultivation and human influence (highest accuracies for Uganda GH event, lowest for Rwanda GH
event). The magnitude of the seasonal vegetation oscillations, which shows connectivity with the
precipitation patterns (fig. 6), varies significantly with changing landscapes and results in profound
seasonal cyclicity in both the amplitude and coherence timeseries. Although the coherence is additionally
influenced by atmospheric effects (Rocca et al., 2000), the influence of both the vegetation and
atmosphere on the coherence does not obscure the GH event induced change within the time series.
Notably, after detrending, the effects of both seem to be almost negligible. Denser and taller vegetation,
result in lower seasonal cyclicity within the amplitude and coherence time series. S1 operates in C-band
frequency, meaning that the emitted signal penetrates the canopy layer and subsequently bounces on
the branches, and leaves underneath (Dzurisin, 2006). A reduction in vegetation after a seasonal dry
period within sparsely vegetated areas, i.e., the grass- and croplands in the DRC GH event, will likely



expose the soil underneath and have a pronounced influence on the backscattering signal given the
difference in backscattering properties of vegetation and soil (Strozzi et al., 2000, Weydahl, 2001,
Colsesanti & Wasowski, 2006, Tessari et al., 2017). In contrast, a seasonal dry period in a dense forest,
(i.e., Uganda GH event) would affect the density of the canopy cover. However due to the height and
close vicinity of the vegetation to each other a dry period does not necessarily lead to more soil
exposure.  This is  corroborated by  the fact that the NDVI does not change much for the Uganda GH
event, despite the seasonal patterns in precipitation (fig. 6). The regions that are covered with the
denser and most uniform vegetation are commonly environments with the lowest chance of getting
timing information from other sources (media, citizen-observer networks) as compared to GH events in
more inhabited landscapes (Jacobs et al., 2019, Monsieurs et al., 2019,).
The complex reaction of the SAR signal to soil moisture and roughness change can causes both an in-
and decrease of the amplitude at the same GH event location (Mondini et al., 2021, Burrows et al.,
2022). Next to the seasonal influence (fig. 6), this can also be a potential reason why no significant
changes at the timing of the GH event are distinguished for all GH events. The inter-pixel variation
captured in SAC proves to be a good tool to account for both this potential in- and decrease and any
seasonal variation in amplitude values at the location of the GH event and increased timing estimation
accuracy.
The pre-event, co-event, and post-event coherence values of our four GH events correspond with the
study of Tzouvaras et al. (2020), where a distinct difference in pre- (low) and post- (high) GH event
coherence values is observed at the location of a landslide occurrence. We observe the same patterns
with the GH events that contain flash floods, likely because a clear landscape change is observed after
the occurrence of the (often sediment-rich) flash floods (fig. 1). The co-event coherence drop as
observed by Tzouvaras et al. (2020) and Burrows et al. (2019) at the location of a landslide occurrence
does not prove to be significant enough to be able to determine GH event timing. This is most likely
attributed to the fact that the GH events occurred in low-coherence (vegetated) areas (Weydahl 2001,
Tessari et al., 2017).

### 5.1.2 GH event distribution

An increase in GH area improves the accuracy of timing detection, which can likely be related to the
increased number of pixels fully covering the GH feature relative to the fuzzy edge pixels (e.g. Foody
and Mathur, 2006, Deijns et al., 2020, Zhong et al., 2021).
Generally, accuracy is not correlated with slope angle (fig. 9). However, an increase in accuracy with
increasing slope with a relative low reliability is observed for coherence. Nevertheless, this trend must
be considered with a certain caution: (1) the trend is dependent on the quality of terrain correction
during the pre-processing step (section 3.1), which should make SAR values independent of slope angle
(Small, 2011), (2) a changing slope angle could influence the GH size (Chen et al., 2016), (3) we take





the average slope angle per GH. Elongated GH features (mainly the flash flood features in the GH
inventories) will have an average slope angle that is not representative for every part of the GH.
Although a clear difference can be observed in time series response to GH events located in different
landscapes (fig. 6), the comparison with the land cover does not allow to find a clear relationship with
the type of vegetation (fig. 9). Since the land cover distribution is not equal amongst GH events (fig.
5), the results are, for some GH events, based on a low amount of individual GH features, which might
not be representative enough for a general trend. The observed large variation in values per box plot
(fig. 9m-o) might be an indication of this.
By using the right-looking S1 satellite data, foreshortening, and layover effects should be limited with
descending track acquisitions for GH exposed towards the west (180-360°) and with ascending track
acquisitions for GH exposed towards the east (0-180°). The shadow affects in the opposite direction
and is dependent on the slope of the terrain (Bamler, 2000). We see that, generally, the individual GH-
features on the descending slope tend to have a higher timing estimation accuracy for the west facing
slopes and the individual GH features on de ascending track for the east facing slopes, which is as
expected. However, there remains variability in the result, for example, an opposite pattern is visible
for Uganda GH event with the coherence and a partial favorability for the descending track acquisition
on east facing slopes is visible for the DRC GH event. Future research on the detailed effect of changing
GH feature aspects on the ascending and descending SAR time series can provide additional valuable
information in this context.
Our derived trends are established from GH events with each 318 to 1063 individual GH features and
provide a good indication of the SAR response to changing landscape parameters. It remains interesting
to see if these trends sustain with the addition of more GH events from different landscapes.
**5.2. Result considering recent developments in SAR timing detection**
Our results are somewhat in contrast with Burrows et al., (2022), who argue that coherence is less
performant than amplitude for GH event timing. Burrows et al. (2022) used a method similar to our
timing workflow 2, where they estimated timing from individual landslides and chose the majority to
represent the timing. Using amplitude data, they were able to estimate the timing of ∼ 20% of landslides
per inventory with an accuracy of 6-12 days with ~80% confidence. Whereas by using coherence
(60x60m resolution) they acquired much lower confidence values (24-47%). Their study, however,
differs in several aspects from our analysis:
1.  Burrows et al., (2022) applied their method with a pre-defined notion of GH event timing, i.e.
known year and season. For our GH events, we see distinct seasonal dynamics mainly within
the amplitude time series. Zooming in on a specific time frame (3 months before and 3 months
after the GH event occurrence like Burrows et al., (2022)) reduces the overall seasonal
dynamics, which could be the cause of a wrongly identified GH event change. This improves
the detectability of the GH event within the time series and the resulting accuracies. We define



a methodology that requires no knowledge on GH event timing before application, which is an
advantage if no GH event timing is present, however, this increases the chance of any seasonal
influence visible within the time series.
2. They applied their method on landslides only. In our case, the addition of flash floods to the
inventories introduces different types of contrasting GH shapes, slopes and land cover (flash
floods tend to be elongated, occur in the valleys with shallower terrain, whereas landslides
occurred mainly on the steeper hillslopes) that can influence the SAR time series, specifically if
the flash flood enters urbanized area (such as in the DRC GH event) or run through a seasonally
dynamic channel with seasonally changing soil moisture levels influencing the SAR signal (Ulaby
et al., 1996, Scott et al., 2017).
3. Their used landslide inventories (from Roback et al., 2018, Emberson et al., 2022) suggest that
they developed their method on landslides in comparatively more ideal homogenous landscape
conditions, where the landslides generally occurred in denser vegetated areas with less
cultivated area. This is corroborated by their high pre-event NDVI values (peak of distribution
between 0.7-0.8). In the three areas they studied, Zimbabwe, located in a semi-arid climate
region (Roback et al., 2018) is the region where the landscape is the least homogenous and
the closest to what we study. In Zimbabwe however, landslides generally still occur in vegetated
areas (including grassland, forests) without significant agricultural practices, which is in contrast
with the Rwanda and DRC GH events in our study that have a large portion of the GH within
crop- and grassland (fig. 5). In agreement with Burrows et al. (2022) our results show that the
Uganda GH event, where most of the landscape consists of dense vegetation (i.e., the highest
NDVI values), show estimated GH event timing accuracies that are the highest among all GH
events, obtaining a 1-2 images difference from the actual GH event occurrence for SAC (1-16
days) and (detrended) coherence (1-8 days). Although amplitude is overall less performant for
the Uganda GH event, we still achieve an accuracy off 13-22 days for the detrended amplitude.
4. We do not threshold on individual GH area. Specifically, the Rwanda GH event contains a GH
event size distribution that includes many small individual GH features below this threshold (fig.
5). Together with the complexity and large fraction of cultivation of the landscape this clearly
results in reduced estimation accuracies.
5. To improve timing accuracy, they removed timing estimations that did not pass a threshold
based on the relative magnitude of the change in the SAR time series induced by the landslides

**5.3 Improvements and perspectives**

The current methodology successfully allows to analyze GH event timing from SAR, but several
improvements can be considered in future research.


### 5.3.1 Improvements

1. In the current methodology we do not detrend individual GH feature time series. Because detrending does increase timing accuracy within our study, further research on accurate detrending of individual GH time series can potentially greatly benefit timing estimation accuracy.

2. We use one point change detection algorithm (ruptures: Truong et al., 2020) to find changes related to the GH event occurrence within the time series. Comparing multiple change detection algorithms (e.g., the ones used by Burrows et al. (2022)), could potentially benefit GH event timing estimation accuracy. Additionally, within our timing workflow 2, we do not incorporate any methodology to filter out any low accuracy timing estimations, such as in Burrows et al., 2022. However, since we do not apply the methodology with a pre-defined notion of time, our time series are prone to seasonal dynamics (fig. 6) and the applicability of such an implementation remains to be investigated.

3. The quality of the amplitude and coherence imagery is dependent on the quality of the pre-processing applied with the MasTer tool (Derauw et al., 2020, d'Oreye et al., 2021) and how it deals with the different steps such as co-registration, radiometric terrain correction and geocoding. Quality of the imagery in its turn is also dependent on, among others, the multi-look factor (amplitude), the interferometric multi-look factor and the maximum temporal and perpendicular baselines (coherence). In addition, different polarizations may yield different results (Shibayama et al., 2015, Psomiadis, 2016, Park & Lee, 2019) and the use of a different polarization can potentially improve event detectability within the time series. Improvements within the SAR imagery might be achieved by tweaking and closely investigating different pre-processing steps to achieve better image quality.

4. The SAC result depends on the ability to find the best reference image (Section 3.2). Additional efforts can be made to better find the SAC time series that shows the most significant change related to the GH event occurrence. For example, a preliminary filtering of very noisy SAC time series (before applying our developed method using the ECDF's) can potentially benefit the ability to acquire the best reference image.

### 5.3.2 Perspectives

1. We have studied, for the first time in a GH event timing detection approach, both landslides and flash floods in a combined methodology. Since these GH often co-occur and interact (Marengo & Alves, 2012, Jacobs et al., 2016a, Rengers et al., 2016) they should be analyzed in a multi-hazard approach. Our method can be well applied within a multi-hazard methodology. For example, multi-hazard inventories can serve as an input for our methodology to improve event timing accuracy. Regional results can subsequently be used in hazard assessment, early warning, and disaster risk reduction strategies.

2. Our study shows that there is a clear advantage to analyzing different S1 SAR data products when estimating GH event timing. The fact that Burrows et al. (2022) shows better results for amplitude compared to coherence data is in contrast with our results but reinforces the idea of investigating both data products when applying the method to new regions.


3. Regarding transferability of our developed method. Given the clear influence of landscape and climate as controlling factors for SAR time series behavior (elaborated in section 2.3), we aimed to develop our method within a variety of contrasting landscapes and contrasting vegetation dynamics. Slope angle does not seem to influence accuracy (fig. 9). Transferability to other regions seems therefore likely to acquire good results, specifically for the coherence and detrended coherence time series as they do seem less influenced by seasonal dynamics than the amplitude time series. The precipitation regime within our study area is quite similar for the four studied GH events (Nicholson 2017, Monsieurs et al., 2018a). Since soil moisture and wetness have an influence on amplitude and coherence time series (Ulaby et al., 1996, Srivastava et al., 2006, Brancato et al., 2017, Scott et al., 2017), contrasting precipitation regimes within other regions could potentially influence the response of the SAR time series and the estimated GH event timing accuracy. Examples of contrasting precipitation regimes: (1) a lower amount of precipitation in more arid regions, or lower/higher amounts in other tropical regions (Fick and Hijmans, 2017). (2) a change in precipitation seasonal variability due to spatially different oscillation of the ITCZ (Nicholson et al., 2017, Dewitte et al., 2022). (3) the effect of local topography and the presence of lakes on the local precipitation patterns (e.g. Thiery et al., 2015; 2016; 2017, Monsieurs et al., 2018b). The influences of these contrasting precipitation regimes on SAR-based GH timing detection however, remains to be investigated. Additionally, in its current form, the methodology does not account for the GH events that occur within a time span that is longer than the acquisition time (> 6-12 days) of S1 images (i.e. multi-temporal GH events). In that case one would require a time window of occurrence, rather than a specific date. The methodology can be adapted to allow it to derive a time window of GH occurrence. This could mainly be done following timing workflow 1 (the GH event scale). The start and the end date of the GH event inducing change within the SAR time series (applicable for all data products) should be indicative of the time window of GH event occurrence. However, this remains to be investigated.

4. The open-access S1 satellite with its high resolution, high repeat time and global coverage proves to be an excellent data product for estimating GH event timing and allows for our developed method to be applied on every region of the world. The use of our method with different satellite products (e.g. COSMO-SkyMed, upcoming NISAR satellite) is not straightforward. Different available SAR satellite products operate in different bands (X-band for COSMO-Skymed, L-band for NISAR), which have varying vegetation penetration depths (Dzurisin, 2006). The effect related to varying vegetation penetration depths remains to be investigated

5. The method can benefit (in terms of data availability, scalability, and processing time) from implementation on a cloud computing service. However, these cloud computing platforms only provide pre-processed amplitude imagery (i.e. amplitude ground range detected imagery). This will allow for the applicability of our method using the amplitude, detrended amplitude and SAC data products, but there is so far no possibility for processing and using coherence data. Additionally, the use of pre-processed amplitude imagery restrains us from manual input during the pre-processing step (as the MasTer Toolbox allows).

6.  The method can potentially be combined with optical data (e.g. Deijns et al., 2020) that could serve
as additional data to help narrow down the time window and filter out any non-sense timing
estimations.

## 6. Conclusion

We established a new method to automatically determine GH event timing from SAR images, that can
be applied without prior knowledge of the GH event. Our method is original as it is the first time that
landslides and flash floods are studied together. By showing that these two processes can be detected
and therefore studied together, we open new perspectives in the study of multi-hazards, which can aid
in hazard assessment, early warning, and disaster risk reduction strategies. Our methodology has the
potential to be combined with existing spatial detection methods to support inventory creation and boost
GH event research in remote inaccessible areas such as the African cloud-covered tropics.
From a data processing point of view, the method is established around an unprecedented analysis of
various SAR products coming from Sentinel-1 (S1) images. We show that there is a need to investigate
different SAR data products when estimating GH event timing (amplitude, spatial amplitude correlation,
and coherence) since the signal response can be different and sometimes contradictory when looking
at one single event. The implementation of our method on a cloud computing platform can be beneficial
in terms of scalability, data availability and processing time. However, the main limitations in this context
are: (1) no control in pre-processing of S1 imagery and, (2) S1 coherence data is so far not available
within these platforms.
With a focus on four events containing a total of about 2500 landslides and flash flood features in
contrasting landscapes, we propose a method that is adapted to be applied to other regions. Here, we
focused on tropical environments where climate conditions and land use dynamics are rather specific.
However, we believe that the complexity of these landscapes is an added value for the transferability
of the method. Additionally, the use of the globally available open access S1 satellite data allows our
method to be applied on every region of the World.

## Acknowledgement

This study was supported by the Belgium Science Policy (BELSPO) through the PAStECA project
(BR/165/A3/PASTECA) entitled "Historical Aerial Photographs and Archives to Assess Environmental
Changes in Central Africa" (http://pasteca.africamuseum.be/, last access 09 June 2022). The
compilation of the inventory data benefited from field-based insight and discussion with Arthur Depicker,
Josué Mugisho Bachinyaga, John Sekajugo and Judith Uwihirwe. PlanetScope data provided by the
European Space Agency.


**Code and data availability**


Sentinel-1 and Sentinel-2 data are provided open-access by the European Space Agency. Landsat 8 data
are provided open access by the U.S. Geological Survey. The Python scripts for the GH event timing
estimation, sensitivity analysis, and precipitation analysis and the Google Earth Engine code for
vegetation analysis will be provided once the manuscript is accepted for publication.
**Author contribution**
AAJD, OD, FK and WT conceived the study. AAJD compiled the landslide and flash flood inventory with
the support of OD. AAJD processed and analyzed the data. OD conducted field work for the validation
of the inventory. AAJD wrote the original draft of the manuscript, with key initial input from OD and FK.
NO trained AAJD in SAR image pre-processing. All the authors contributed to reviewing and editing the
manuscript. OD obtained funding for this work.
**Competing interests.**
The authors declare no conflict of interest.

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
