# Peer review of "Timing landslide and flash flood events from SAR satellite: a regionally applicable"

_Natural Hazards and Earth System Sciences, 2022_

## Author Response (AR1)

**Referee #1**

**Comment 1:** I believe the strength of the paper relies in the accurate comparison of those approaches, rather than in the new proposed approach. I'd change the tile accordingly, as well as underline this in the text.

*Initial Reply*: One of the strengths of the paper is indeed in the use of a number of SAR data products that we deliberately tested within contrasting landscapes. This clearly allowed us to better understand the ability of different SAR products for detecting the timing of these interacting landslide and flash flood events within different landscape conditions and will help us when applying our methodology on a larger set of events. From an event timing detection perspective, however, we are the first one who use SAR to analyze landslides and flash floods together as being co-occurring and interacting events. This combination of geomorphic hazards is quite frequently leading to societal and environmental impacts that are more severe. However, such processes are usually studied in isolation, then leading to an underestimation of their impacts. One key step to study these combined processes together is to collect information on their temporal occurrence. However, these processes are almost never studied together and, so far, there has never been a research dedicated to their combined temporal detection using radar satellite. The use of an unprecedented combination of SAR products, plus the approach of analyzing events containing both co-occurring and interacting landslides and flash floods explains our use of 'a new methodology' (as it is summarized in lines 556-560). In our revised manuscript, we will put more emphasis on the fact that we intentionally process landslides and flash floods together to make it clearer for the reader.

*Authors reply:* As said in our initial reply, there is no literature, on a well-defined methodology to accurately estimate interacting and co-occurring landslide and flash flood event timing using SAR. The recent work of Burrows et al, 2020 is the research that comes the closest to our work. However, they focus only on landslide timing estimation. On top of that, the methodology of Burrows et al., 2022 only focus on highly vegetated areas (which they mention themselves in the manuscript often is the ideal location for GH event timing detection). Therefore, their methodology is potentially not fit for different types of landscapes. A new regionally applicable methodology addressing this issue of co-occurring GH event is therefore a necessity. Our results do in fact show that it is very important to take the complex nature of the landscape into account when developing a method, as amplitude results in highly cultivated urbanized areas posed to be much less reliant than in highly vegetated areas. This clearly showed that a better-defined methodology applicable on the regional scale should be developed. To better highlight the fact that we create this methodology to be usable in a wide variety of landscapes, instead of focusing on the novel aspect, we focus on the regional applicability aspect of the methodology. Hence, we changed the title accordingly.

We have changed the word 'method' to 'methodology' throughout the manuscript, since it is the combination of methods, that describes the novelty of this research.

Thereby, we have made it clearer throughout the manuscript that we study landslides and flash floods together instead of separate processes.

Also, we made it clearer in the introduction and conclusion that we use an unprecedented amount of SAR derived products (amplitude, SAC, coherence), to highlight better highlight this as a strength for the paper

List of relevant changes:

- Line 1-2*: 'new method'* to *'regionally applicable methodology'*
- Line 20: *'regionally applicable methodology'*
- Line 27: *'method'* to *'methodology'*
- Line 43: *'method'* to *'methodology'*
- Line 50-51: Added new lines: *'Landslides and flash floods are often studied in isolation. However, it is their combined occurrence that can lead to more extreme impacts'*

- Line 109-110: Added new lines: *'However, so far, there has never been research dedicated to their combined temporal detection using radar satellite.'*
- Line 116: *'regionally applicable methodology'*
- Line 117-118: *'We analyze landslides and flash floods together as being co-occurring and interacting events.'*
- Line 119: *'method'* to *'methodology'*
- Line 120-122: *"The methodology is developed using four GH events either containing landslides, or a combination of landslides and flash floods located in contrasting landscape types observed within tropical Africa (see section 2.1)."*
- Line 122-123: *'We analyze an unprecedented amount of S1 SAR products, namely:'*
- Line 331: *'method'* to *'methodology'*
- Line 611: *'regionally applicable'*
- Line 611: *'regionally applicable'*
- Line 615: *'method'* to *'methodology'*
- Line 618: remove *'new method'* + add *'methodology'*
- Line 784: *'method'* to *'methodology'*
- Line 791: *'method'* to *'methodology'*
- Line 794: *'method'* to *'methodology'*
- Line 821: *'method'* to *'methodology'*
- Line 831: *'method'* to *'methodology'*
- Line 834: *'method'* to *'methodology'*
- Line 839: *'method'* to *'methodology'*
- Line 843: remove *'new'* + add *'regionally applicable methodology'*
- Line 844-846: added *'We successfully assessed the use of multiple SAR derived data products in their ability to accurately detect GH event timing in contrasting landscapes.'*
- Line 853: *'method'* to *'methodology'*
- Line 857: *'method'* to *'methodology'*
- Line 862: *'method'* to *'methodology'*
- Line 865: *'method'* to *'methodology'*
- Line 866: *'method'* to *'methodology'*

> **Comment 2:** **Authors state that reducing the investigation time frame would increase the accuracy, however they consider that no time information is available, while all the methods require the inventory of the phenomena. Now, in cases in which the inventory of the event is available, and the study cases are multiple GH events, I believe the timing is more or less known (at least with +- 6 months of uncertainty). Why did you decide to make such assumption?**

*Initial Reply:* The methodology proposed in this research is designed to be applied automatically with minimal intervention, with a focus on the regional scale. More specifically, tropical highly cloud covered areas where data scarcity is prevalent are the first target. In large regions, such as in our study area, information on the temporal distribution of GH events may not always be available. We will make sure that it is better understood.

*Authors reply:* Extra explanation and elaboration is added in the discussion at lines 709-713

List of relevant changes:

- Line 709-713: Rewritten: *"Reducing this time window will potentially improve the detectability of the GH event within the time series. However, our methodology is intended to be applicable in areas such as the western branch of the East African Rift, an area characterized by data scarcity (Dewitte et al., 2021). In areas like these, information on the temporal distribution of GH events may not always be available."*

**Comment 3: It is not clear how, after processing, the time of occurrence is set by time series analysis.**

*Initial Reply*: This is addressed in lines 351-358. For this we are using a change detection package named "rupture", that uses binary segmentation to derive the most significant change within the time series. The resulting variable is basically a point in time. In our revised manuscript, we will rephrase to make this clearer.

*Authors reply:* We have elaborated on the application of the python package 'ruptures' that is used to define the date of occurrence of the GH event. Changes made in lines 382-390

List of relevant changes:

- Line 382-390*: Rewritten to: "Timing is defined on every time series (for amplitude, SAC and coherence) using a binary segmentation change detection approach (Bai, 1997, Fryzlewicz, 2014) using the python package 'Ruptures' (Truong et al., 2020) (fig. 3, Step 4). The algorithm was set to predict only one breakpoint since we aim to detect the most significant change in the time series. The output of the applied binary segmentation change detection algorithm is a value that represents the location of the image within the image stack. The date of this image is extracted and Assigned as/ the earliest date after the GH event occurrence. This applies for the amplitude and SAC time series. However, since coherence is based on image pairs, it would identify the image pair right after the GH event. We therefore assign the first date from this image pair as the earliest date after the GH event occurrence. "*

**Comment 4: Coordinates missing in Figure 1. This figure must be improved.**

*Initial Reply:* Agree, the coordinates will be added in the next iteration.

*Authors reply:* Added coordinates in figure 1 at line 166. Also added numbers per GH event so that they correspond with the overview image in the bottom left + changed the north arrow

List of relevant changes:

- Line 166: New version of figure 1 added

**Comment 5: Put coordinates outside each tile of figure 2 instead than into caption.**

*Initial Reply:* -

*Authors reply:* Agree, coordinates are added in figure 2 at line 193.

List of relevant changes:

- Line 193 New version of figure 2 added
- Line 194-196: Deleted coordinates in the caption. Changed the copyright of figure 2a (to *Google, ©2022 CNES/Airbus*).

**Comment 6: Figures 2, 7 and 8 could be improved.**

*Initial Reply:* -

*Authors reply:* Figure 2, line 193, Uganda and Burundi tile are different images. They are now less dark. To us, figure 7 and figure 8 are understandable. We would have welcomed any suggestions. We have changed figure 7 and 8 to match the colors assigned to amplitude (red), SAC(green) and coherence (black) in the other figures (6 & 9). Additionally figure 7 and 8 are changed so that they better match the layout of figure 6.

List of relevant changes:

- Line 193: New version of figure 2 added
- Line 194-196: Deleted coordinates in the caption. Changed the copyright of figure 2a (to *Google, ©2022 CNES/Airbus*).
- Line 547: New version of figure 7 added

- Line 548-555: improved on the caption in figure 7 and matched better with the new caption in figure 8: *"Estimated GH event timing using the complete GH event scale (workflow 1) for amplitude, detrended amplitude (red), SAC (green), coherence and detrended coherence (black). The darker colored bar representing the ascending track results. The lighter colored bar representing the descending track results. The length of bars represent the uncertainty in timing (see section 3.3). Dashed lines on the bars represent the overlap between the ascending and descending track results."*
- Line 557: New version of figure 8 added
- Line 558-564: Changed caption to better match caption from figure 7: *"Estimated GH event timing using the complete GH event scale (workflow 1) for amplitude, detrended amplitude (red), SAC (green), coherence and detrended coherence (black). The darker colored bar representing the ascending track results. The lighter colored bar representing the descending track results. The length of bars represent the uncertainty in timing (see section 3.3). Dashed lines on the bars represent the overlap between the ascending and descending track results."*

**Comment 7:** **I would condense the text. This would make it easy for readers to follow the manuscript flow. Sometimes the same details are repeated several times in the text.**

*Initial Reply:* We will make sure that repetition in details is reduced to improve the quality of the text; noting nevertheless that reviewers #1 and #2 both praised the quality of the writing. Powered by

*Authors reply:* We have tried to condense and adapt the text throughout the manuscript to improve the readability and flow of the text. Also, some typos were addressed accordingly. We specifically made some name changes in section 3 (methods) to avoid any confusion due to naming. Below are the most substantial changes made.

List of relevant changes:

- Line 29-34: rephrased part of the abstract to a more interpretable that is a similar style to a similar part in the discussion (Line 621-625): *"The amplitude and detrended amplitude time series in our methodology do not prove to be effective in accurate GH event timing estimation, with estimated timing accuracies ranging from a 13 day to a 1000 days difference. A clear increase in accuracy is obtained from SAC with estimated timing accuracies ranging from a 1 day to an 85 day difference. However, the most accurate results are achieved with coherence and detrended coherence with estimated timing accuracies ranging from a 1 day to a 47 day difference."*
- Line 120-122: added some text to increase flow and readability: *"The methodology is developed using four GH events either containing landslides, or a combination of landslides and flash floods located in contrasting landscape types observed within tropical Africa (see section 2.1). We analyze an unprecedented amount of S1 SAR products, namely:"*
- Line 321: Change to figure 3: *'GH time series'* to *'Amplitude & Coherence'* to make it match better with step 2 *'Spatial Amplitude Correlation'*.
- Line 322: *'Workflow'* to *'Flowchart'* to make a better distinction to the flowchart presented in figure 3 and the workflows described in 344-352.
- Line 344-352: New layout of the text, plus additional text to explain the workflow and flowchart difference
- Line 345-346: added *"In this workflow, the steps outlined in figure 3 are carried out once using all pixels encompassing the full GH event"*
- Line 349-350: added *"and the steps outlined in figure 3 are carried out separately for each individual GH feature"*
- Line 358: fig 3.3 changed to: *"fig3, step 3"*
- Line 359: fig 3.2 changed to: *"fig3, step 2"*
- Line 369: *"section"* to *"step"*
- Line 376-381: rewriting
- Line 398-402: Removal of sentences that are better incorporated in lines 388-390

- Line 415: rephrased section head to better match the content: *"Sensitivity analysis with respect to landscape characteristics"*
- Line 470 – 472: removal and better incorporated at 466-467
- Line 493 – 499: removal and rewriting as this was already explained in 402-404
- Line 512-514: Removal, already explained in 488-494
- Line 698 – 700: Removal, unnecessary information
- Line 724-733: Removal, unnecessary information. This part is now better specified in Burrows et al 2022 after publishing, so no need for large portion of this text.
- Line 759-763: Removal, this point is already mentioned in lines 744-747

**Referee #2**

**Comment 1:** **I do not believe that the use of Time Series using SAR data to estimate the timing of an event can be considered a novel practice. Several attempts have been performed either using the phase, thus the displacement itself (see Intrieri et al., 2018), and Burrows, as correctly written in the manuscript. Besides, the methodologies here depicted are based on standard change detection based on the trends.**

*Initial Reply*: To answer this concern, we would like to reiterate an answer provided to reviewer #1 as part of our reply. One of the strengths of the paper is in the use of a number of SAR data products that we deliberately tested within contrasting landscapes. This clearly allowed us to better understand the ability of different SAR products for detecting the timing of these interacting landslide and flash flood events within different landscape conditions and will help us when applying our methodology on a larger set of events. From an event timing detection perspective, however, we are the first one who use SAR to analyze landslides and flash floods together as being co-occurring and interacting events. This combination of geomorphic hazards is quite frequently leading to societal and environmental impacts that are more severe. However, such processes are usually studied in isolation, then leading to an underestimation of their impacts. One key step to study these combined processes together is to collect information on their temporal occurrence. However, these processes are almost never studied together and, so far, there has never been research dedicated to their combined temporal detection using radar satellite. The use of an unprecedented combination of SAR products, plus the approach of analyzing events containing both co-occurring and interacting landslides and flash floods explains our use of 'a new methodology' (as it is summarized in lines 556-560). In our revised manuscript, we will put more emphasis on the fact that we intentionally process landslides and flash floods together to make it clearer for the reader. For the inserted reference, I assume you are referring to this one:

Intrieri, E., Raspini, F., Fumagalli, A., Lu, P., Del Conte, S., Farina, P., Allievi, J., Ferretti, A. and Casagli, N., 2018. The Maoxian landslide as seen from space: detecting precursors of failure with Sentinel-1 data. Landslides, 15(1), pp.123-133.

In regard to this reference, I would like to highlight that we are not using ground deformation. And that using a deformation approach would be impossible given the high velocities these GH events have (shallow landslides and flash floods). Also, for each GH event we see very low coherence values that additionally provide constraints to using ground deformation. Burrows et al. (2022) have been developing a methodology to detect landslide timing using SAR, but specifically only using the amplitude product. We analyze an unprecedented combination of SAR products within contrasting landscapes for timing detection of co-occurring and interacting landslide and flash flood events. So, we therefore consider it a novel methodology.

Burrows, K., Marc, O., and Remy, D.: Establishing the timings of individual rainfall triggered landslides using Sentinel-1 satellite radar data, Nat. Hazards Earth Syst. Sci. Discuss. [preprint], https://doi.org/10.5194/nhess-2022-21, in review, 2022.

Authors reply: As said in our initial reply, there is no literature, on a well-defined methodology to accurately estimate interacting and co-occurring landslide and flash flood event timing using SAR. The recent work of Burrows et al, 2020 is the research that comes the closest to our work. However, they focus only on landslide timing estimation. On top of that, the methodology of Burrows et al., 2022 only focus on highly vegetated areas (which they mention themselves in the manuscript often is the ideal location for GH event timing detection). Therefore, their methodology is potentially not fit for different types of landscapes. A new regionally applicable methodology addressing this issue of co-occurring GH event is therefore a necessity. Our results do in fact show that it is very important to take the complex nature of the landscape into account when developing a method, as amplitude results in highly cultivated urbanized areas posed to be much less reliant than in highly vegetated areas. This clearly showed that a better-defined methodology applicable on the regional scale should be developed. To better highlight the fact that we create this methodology to be usable in a wide variety of landscapes, instead of focusing on the novel aspect, we focus on the regional applicability aspect of the methodology. Hence, we changed the title accordingly.

We have changed the word *'method'* to *'methodology'* throughout the manuscript, since it is the combination of methods, that describes the novelty of this research.

We made it clearer in the introduction and conclusion that we use an unprecedented amount of SAR derived products (amplitude, SAC, coherence) to highlight this part of the manuscript better.

Additionally, we do not believe that Intrieri et al., 2018 intended to use deformation to identify the timing within the time series. As they clearly stated in the paper, they intend to identify precursor movements as a means to use them for landslide forecasting. In its current state, the methodology of Intrieri et al., 2018 does not show any indication of ability to identify the timing when the GH event timing is not known. We therefore did not consider it essential literature on GH event timing detection that should be mentioned at the same level of Burrows et al. (2022). As a note, Burrows et al., 2022, do not consider deformation studies in their publication either. However, we have improved the introduction by adding some literature on the deformation studies, to better put our research into perspective with those studies.

Also, we have changed the text to apply on the most recent version of Burrows et al., 2022. Changes detailed below.

List of relevant changes:

- Line 1-2: *'new method'* to *'regionally applicable methodology'*
- Line 20: *'regionally applicable methodology'*
- Line 27: '*method'* to *'methodology'*
- Line 43: *'method'* to *'methodology'*
- Line 86-91: added *"Additionally, there is a wide range of studies that use SAR-derived ground deformation to map landslides (Casagli et al., 2017, Solari et al., 2020) or analyze pre-cursor movements (Intrieri et al., 2018) and internal variability (Nobile et al., 2018). However, they are dependent on consistent high coherence values at the GH locations, which will make these methods of limited use in highly vegetated landscapes (e.g. the tropics) (Komac et al., 2015; Solari et al., 2020) and for fast moving GH (e.g. shallow landslides and flash floods) (Burrows et al., 2020; Tzouvaras et al., 2020)."*
- Line 103-104: Removed, since Burrows et al., 2022 uses detrending therefore they do consider vegetation dynamics within some of their detection techniques.: *"and (5) without consideration of the effect of vegetation dynamics within the timespan."*
- Line 116: *'regionally applicable methodology'*
- Line 119: *'method'* to *'methodology'*

- Line 122-123: added *'We analyze an unprecedented amount of S1 SAR products, namely:'*
- Line 331*: 'method' to 'methodology'*
- Line 611: *'regionally applicable'*
- Line 615: *'method' to 'methodology'*
- Line 618: remove *'new method'* + add *'methodology'*
- Line 724-733: Removal, unnecessary information. This part is now better specified in Burrows et al 2022 after publishing, so no need for large portion of this text.
- Line 784: *'method' to 'methodology'*
- Line 791: *'method' to 'methodology'*
- Line 794: *'method' to 'methodology'*
- Line 821: *'method' to 'methodology'*
- Line 831: *'method' to 'methodology'*
- Line 834: *'method' to 'methodology'*
- Line 839: *'method' to 'methodology'*
- Line 843: remove *'new'* + add *'regionally applicable methodology'*
- Line 844-846: added *'We successfully assessed the use of multiple SAR derived data products in their ability to accurately detect GH event timing in contrasting landscapes.'*
- Line 853: *'method' to 'methodology'*
- Line 857: *'method' to 'methodology'*
- Line 862: *'method' to 'methodology'*
- Line 865: *'method' to 'methodology'*
- Line 866: *'method' to 'methodology'*

**Comment 2.** **Another aspect which should be clarified is related to the trend change threshold. Is there any quantitatively and standardized measure of the change which could be defined for each time series? How can be discriminated a change in the amplitude or coherence trends over the time? Is there a numerical thresholding? If so, how this is calculated, and this can be exported?**

*Initial Reply:* The same issue is raised by reviewer #1. The explanation on this is addressed in lines 351-358. To derive the timing, we use a change detection package "rupture", that uses binary segmentation to derive the most significant change within the time series. The resulting variable is basically a point in time. In our revised manuscript, we will rephrase and expand a bit more on the binary segmentation approach to make the methodology clearer

Authors reply*:* We have elaborated on the application of the python package 'ruptures' that is used to define the date of occurrence of the GH event. Changes made in lines 382-390

List of relevant changes:

- Line 382-390: rewritten and added: *'Timing is defined on every time series (for amplitude, SAC and coherence) using a binary segmentation change detection approach (Bai, 1997, Fryzlewicz, 2014) using the python package 'Ruptures' (Truong et al., 2020) (fig. 3, Step 4). The algorithm was set to predict only one breakpoint since we aim to detect the most significant change in the time series. The output of the applied binary segmentation change detection algorithm is a value that represents the location of the image within the image stack. The date of this image is extracted and assigned as the earliest date after the GH event occurrence. This applies for the amplitude and SAC time series. However, since coherence is based on image pairs, it would identify the image pair right after the GH event. We therefore assign the first date from this image pair as the earliest date after the GH event occurrence.'*

**Comment 3.** **I have also some concerns about the reference to both landslides and flaash flood. Do they behave at the same way in terms of amplitude and coherence? Results show very different timing detection results, however, from the time series analysis seems that this is mostly due to the size of the event. Is there any implication also considering the differences between flash floods and**

**landslides (I would rather consider smaller flash floods since the source and the travel areas can be considered limited with respect to landslides.**

*Initial Reply:* Thanks for addressing this issue. We would like to emphasize that we intentionally analyze landslides and flash floods as combined processes; hence the terminology GH events. Often, these landslides and flash floods co-occur and interact leading to more severe impacts (highlighted in line 41-48). We are interested in these GH events as a whole. We intend to use our developed methodology to automatically identify the timing of these GH events in contrasting landscapes, with a high uncertainty in timing (cloud-covered data scarce tropics). We are therefore not interested in their individual parts. This separate analysis of the landslides and flash floods within these GH events is therefore out of scope. We indeed believe that this combined aspect of the analysis must be better emphasized. In our revised manuscript, we will put more emphasis on the fact that we intentionally process landslides and flash floods together to make it clearer for the reader.

Authors reply*:* We have made it clearer in the introduction that we aim to analyze both landslides and flash floods in a combined methodology, rather than studying them in isolation.

List of relevant changes:

- Line 50-51: Added new lines: *'Landslides and flash floods are often studied in isolation. However, it is their combined occurrence that can lead to more extreme impacts'*
- Line 109-110: Added new lines: *'However, so far, there has never been research dedicated to their combined temporal detection using radar satellite.'*
  Line 117-118: 'We analyze landslides and flash floods together as being co-occurring and interacting events.

**Comment 4.** **Besides, I see a very poor relationship with rainfalls which can be considered a very significant factor for the triggering of such events. Would you please provide a comment?**

*Initial Reply:* We added the monthly cumulative rainfall and the NDVI time series in order to better understand the seasonal cyclicity within the SAR data products time series. Process understanding is out of the scope of this manuscript. Note, that we can add that a spike in monthly cumulative rainfall time series is not necessarily found at the time of the GH event. First, peaks within daily cumulative rainfall do not necessarily lead to peaks within monthly cumulative rainfall, and second, the spatial resolution of our used satellite rainfall products is sometimes too coarse to detect local convective rainfalls that are associated with the GH events. See for example the works that have been carried out in our study area.

Monsieurs, E., 2020. The potential of satellite-rainfall estimates in assessing landslide hazard in Tropical Africa. Royal Museum for Central Africa and University of Liège PhD thesis.

Monsieurs, E., Kirschbaum, D.B., Tan, J., Maki Mateso, J.-C., Jacobs, L., Plisnier, P.-D., Thiery, W., Umutoni, A., Musoni, D., Mugaruka Bibentyo, T., Bamulezi Ganza, G., Ilombe Mawe, G., Bagalwa, L., Kankurize, C., Michellier, C., Stanley, T., Kervyn, F., Kervyn, M., Demoulin, A., Dewitte, O., 2018. Evaluating TMPA Rainfall over the Sparsely Gauged East African Rift. Journal of Hydrometeorology 19, 1507–1528. doi:10.1175/JHM-D-18-0103.1

Nakulopa, F., Vanderkelen, I., Van de Walle, J., van Lipzig, N.P.M., Tabari, H., Jacobs, L., Tweheyo, C., Dewitte, O., Thiery, W., 2022. Evaluation of High-Resolution Precipitation Products over the Rwenzori Mountains (Uganda). Journal of Hydrometeorology 23, 747–768. doi:10.1175/jhm-d-21-0106.1

Authors reply*:* We argued that this comment is out of the scope of our manuscript. Therefore, no changes were made regarding this comment.

List of relevant changes: -

**Comment 5: The introduction section is well written and addresses correctly the scientific theme and the missing gaps. I would just improve and provide more detail in the brief description of the paper and its main outlines in the very end of the introduction.**

*Initial reply:*

Authors reply*: In the brief description of the paper (lines 116-128),* we have elaborated so that it is better understood that we are studying co-occurring events as a whole and we are not interested in their separate parts. We additionally elaborated a bit on the GH events that we are using, and better highlighted the fact that we use an unprecedented amount of S1 SAR data products. Now, we think this section clearly outlines the scope and aim of the paper whilst also highlighting the most important parts of this research.

List of relevant changes:

- Line 116: *'regionally applicable methodology'*
- Line 117-118: added: *'We analyze landslides and flash floods together as being co-occurring and interacting events.'*
- Linen120-122*: added and rewritten: "The methodology is developed using four GH events either containing landslides, or a combination of landslides and flash floods located in contrasting landscape types observed within tropical Africa (see section 2.1)."*
- Line 122-123: added: *'We analyze an unprecedented amount of S1 SAR products, namely:'*

**Comment 6: Figure 1 should be redraw also including any reference to flash floods and landslides. Is there any way to distinguish them?**

*Initial reply: -*

Authors reply*:* Partly agree, we have changed the figure so that it contains coordinates. However, as explained earlier, we are not interested in the separate analysis of landslides and flash floods. During our analysis, we did analyze the GH event as a whole and did not make a distinction between landslides and flash floods. Therefore, we do not intend do this in figure 1 either.

List of relevant changes:

- Line 166: New version of figure 1 added

**Comment 7: Would you please provide a table about the S1 database used in the research framework?**

*Initial Reply: -*

Authors reply*: We believe it is not relevant.* As described in lines 201-205. We use between 196 and 208 ascending and 120 to 193 descending images per GH event. Providing a table with hundreds of images would be unreadable and, in our opinion, not necessary. We have added lines in this section to make it more understandable and easier to the users to find the datasets. We made changes in line 201-205 so that it is clear that we make use of the full time series available at the location of the GH events. We included the tracks that we used so that it is easy to find the images for the reader.

List of relevant changes:

- Line 201-205: rewritten*: 'To study the four GH events (fig. 1) we use all available high resolution S1 imagery (~15x15 meter resolution) from January 2016 to January 2021 at the location of the GH event at tracks 174 (ascending) and 21 (descending). This equals to between 196 and 208 ascending and 120 and 193 descending images per GH event, where images occasionally overlap more than one GH event.'*

**Comment 8: Line 251: why did you keep a rectangular cell even using a multilooking factor?**

*Initial Reply: -*

Authors reply: We decided to multi-look the amplitude images to reduce the speckle and smooth the images slightly. In our workflow we decided to transform the images from slant range to ground range to make them compatible with the GH events that we manually delineated. We then chose Master Toolbox to force a pixel size for all imagery to make them consistent through time and space.

List of relevant changes:

- Line: 275: added *', to reduce speckle'*